# On the Algorithmic Stability of Adversarial Training

**Yue Xing**
Department of Statistics
Purdue University
xing49@purdue.edu

**Qifan Song**
Department of Statistics
Purdue University
qfsong@purdue.edu

**Guang Cheng**
Department of Statistics
Purdue University
chengg@purdue.edu

## Abstract

The adversarial training is a popular tool to remedy the vulnerability of deep learning models against adversarial attacks, and there is rich theoretical literature on the training loss of adversarial training algorithms. In contrast, this paper studies the algorithmic stability of a generic adversarial training algorithm, which can further help to establish an upper bound for generalization error. By figuring out the stability upper bound and lower bound, we argue that the non-differentiability issue of adversarial training causes worse algorithmic stability than their natural counterparts. To tackle this problem, we consider a noise injection method. While the non-differentiability problem seriously affects the stability of adversarial training, injecting noise enables the training trajectory to avoid the occurrence of non-differentiability with dominating probability, hence enhancing the stability performance of adversarial training. Our analysis also studies the relation between the algorithm stability and numerical approximation error of adversarial attacks.

## 1 Introduction

Successful machine learning algorithms require not only a good empirical performance but also generalizing well to unseen data. For the robustness towards unseen data, empirical experiments show that deep learning models can be fragile and vulnerable against adversarial input (Biggio et al., 2013; Szegedy et al., 2014). To set an example, in image recognition problems, a deep neural network will predict a wrong label when the testing image is slightly altered, while the change is barely recognizable by human eyes (Papernot et al., 2016a).

Related research efforts in adversarial learning include designing adversarial attacks in various applications (Papernot et al., 2016a,b; Moosavi-Dezfooli et al., 2016), detecting attacked samples (Tao et al., 2018; Ma and Liu, 2019), and modifications on the training process to obtain adversarially robust models, i.e., adversarial training (Shaham et al., 2015; Madry et al., 2017; Jalal et al., 2017).

However, although adversarial training improves the adversarial robustness during testing, its generalization performance is still poor. While Yin et al. (2018) presented that the adversarial Rademacher complexity is never smaller than its natural counterpart, Schmidt et al. (2018); Zhai et al. (2019) argued that a better adversarial generalization requires more labeled/unlabeled data.

In the literature of natural DNN optimization via iterative gradient moves, the empirical loss at each iteration can be characterized by convergence rate analysis, yet generalization properties are not well understood. To characterize the generalization error, one popular way is to study the algorithmic stability. Algorithmic stability is first considered by Kearns and Ron (1999); Bousquet and Elisseeff (2001, 2002). Later, Hardt et al. (2016) explored the connection between algorithmic stability and generalization performance of gradient-type optimization. Some follow-up research studies the stability for different classes of algorithms, or relax the definition of stability to generalize its usage, see Ramezani-Kebrya et al. (2018); Charles and Papailiopoulos (2018); Kuzborskij and Lampert (2018); Zhou et al. (2018); Lei and Ying (2020); Ho et al. (2020); Madden et al. (2020).

35th Conference on Neural Information Processing Systems (NeurIPS 2021).

In general, there are two ways to utilize algorithmic stability. On the one hand, as showed by Hardt et al. (2016), the algorithmic stability provides an upper bound for the generalization error; hence it will be useful when establishing the convergence of generalization error. On the other hand, the algorithmic stability itself is also a measure that evaluates the performance of an algorithm.

Our work extends algorithmic stability analysis to adversarial training. Our contributions are:

- Through figuring out the stability upper bound and lower bound, we argue that adversarial training leads to poor algorithmic stability even the clean loss is smooth. To solve this problem, we propose to inject noise into the adversarial training process. Although some existing works proposed the usage of noise injection, we highlight that such a method is more meaningful in adversarial training than its natural counterpart. Theoretical justification of the noise injection method is provided for a wide range of data-generating models in several tasks, including both linear regression and logistic classification.

- Noticing that, in practice, adversarial attacks are mostly approximated via numerical methods, e.g., fast gradient method (FGM) or projected gradient method (PGD), our theory investigates the role of accuracy of attack approximation for the stability of adversarial training algorithms. In short, a more accurate attack leads to better stability upper bound.

- The effectiveness of noise-injected adversarial training is further generalized to the $\mathcal{L}_\infty$ attack. Compared with $\mathcal{L}_2$, $\mathcal{L}_\infty$ training algorithm is generally less stable.

- Beyond the theoretical analysis under simple models, we provide a theory in two-layer ReLU network with lazy training (training the hidden layer) and observe the effectiveness of the noise injection method. We also obtain empirical evidence that for deep neural networks model, proper forms of noise injection and more accurate attack calculation (e.g., PGD-$k$ over FGM) improve the generalization error.

## 2 Related Works

**Theories in adversarial training**  To theoretically understand how adversarial training works, Sinha et al. (2018); Wang et al. (2019a) investigated the convergence of adversarial training when the loss is strongly convex w.r.t. data attributes. In this case, it can be shown that the gradient of adversarial loss w.r.t. model parameters is Lipschitz, leading to good stability. However, when the loss is not strongly concave in data attributes, Xing et al. (2021a) figured out that adversarial training does not have a Lipschitz gradient even for linear regression. Some studies in deep neural networks (Gao et al., 2019; Zhang et al., 2020; Allen-Zhu and Li, 2020) studied the convergence of adversarial training loss, and Allen-Zhu and Li (2020) also provided a theoretical guarantee of the adversarial testing loss when attack strength is small enough. Some other literature in the generalization of adversarial training can be found in Khim and Loh (2018); Awasthi et al. (2020); Pinot et al. (2021); Xing et al. (2021b).

**Observations in deep learning**  In terms of empirical studies, He et al. (2019); Zheltonozhskii et al. (2020); Xie et al. (2020); Lee and Chandrakasan (2020); Wu et al. (2020) focused on improving the performance of attack/adversarial robustness in deep learning.

In the literature, there are several ways to improve the adversarial training, including modifying the objective function to help the convergence of the training process (Zhang et al., 2019; Wang et al., 2019b), regularization (He et al., 2019; Zheltonozhskii et al., 2020; Wu et al., 2020), replacing non-smooth components (Lee and Chandrakasan, 2020; Xie et al., 2020), and handling over-fitting issue (Lee and Chandrakasan, 2020).

**Stability for non-smooth loss and min-max problem**  Besides works in the algorithmic stability of first-order optimization methods on smooth loss, Bassily et al. (2020) studied the case when loss is convex but not smooth. In this scenario, the minimax lower bound and convergence upper bound of stability together imply that stochastic gradient descent[1] (SGD) and gradient descent (GD) have poor stability. It is recommended to run SGD/GD with an extremely small learning rate for a vast number of iterations, which implies that it is not practical to train a non-smooth model with good stability. Consequently, adaptations are essential for non-smooth models to improve the training process.

---

[1]We are considering sample-with-replacement SGD.

Another recent work, Farnia and Ozdaglar (2020), considered the algorithmic stability in the min-max problems for generative adversarial networks (GAN) to argue that simultaneous training in generator and discriminator leads to good stability. However, besides the strongly-convex-concave assumption in their loss, the "min-max" problem considered in GAN and adversarial training are not the same. These two differences lead to different conclusions between GAN and adversarial training.

# 3 Stability of adversarial training

In this section, we study the uniform argument stability (UAS) of adversarial training. Utilizing the notations introduced in Section 3.1, we present in Section 3.2 the upper and lower bounds of UAS. Section 3.3 studies the effect of attack error on stability.

## 3.1 Notations

**Adversarial training**  To introduce adversarial training, let $l$ denote the loss function and $f_\theta(x)$ be the model with parameter $\theta$. The (population) adversarial loss is defined as

$$R(\theta, \epsilon) := \mathbb{E}\left[l\left(f_\theta[x + A_\epsilon(f_\theta, x, y)], y\right)\right],$$

where $A_\epsilon$ is an attack of strength $\epsilon > 0$ and intends to deteriorate the loss in the following way

$$A_\epsilon(f_\theta, x, y) := \underset{z \in B_p(0, \epsilon)}{\operatorname{argmax}} \{l(f_\theta(x + z), y)\}, \tag{1}$$

where $B_p(x, r)$ is a $\mathcal{L}_p$ ball centering at $x$ with radius $r$.

Given $n$ i.i.d. samples $S = \{(x_i, y_i)\}_{i=1}^n$, the adversarial training minimizes the sample version of $R(\theta, \epsilon)$ w.r.t. $\theta$:

$$R_S(\theta, \epsilon) = \frac{1}{n} \sum_{i=1}^n l\left(f_\theta[x_i + A_\epsilon(f_\theta, x_i, y_i)], y_i\right), \tag{2}$$

and the estimator $\widehat{\theta}$ aims to minimize $R_S(\theta, \epsilon)$. We rewrite $R_S(\theta, \epsilon)$ as $R_S(\theta)$ for simplicity when there is no confusion.

The minimization in (2) is often implemented through an iterative two-step (min-max) update. In the $t$-th iteration, we calculate the adversarial sample $\widetilde{x}_i^{(t)} = x_i + A_\epsilon(f_{\theta^{(t)}}, x_i, y_i)$ based on the current $\theta^{(t)}$, and then update $\theta^{(t+1)}$ based on the gradient of the adversarial training loss while fixing $\widetilde{x}_i^{(t)}$'s with learning rate $\eta_t$. The algorithm runs for $T$ iterations. A more detailed pseudocode is postponed to Algorithm 1 when introducing our adaptations. Note that for some loss function $l$ or model $f_\theta$, there may not be an analytic form for $A_\epsilon$ (e.g. deep neural networks), and numerical methods, e.g. FGM and PGD, are utilized to approximate $A_\epsilon$.

**Risk decomposition**  Define $\theta_0$ and $\bar{\theta}$ as the minimizer of $R$ and $R_S$ respectively. Then for the algorithm output $\widehat{\theta}$, the excess risk can be decomposed into four parts as below:

$$R(\widehat{\theta}) - R(\theta_0) \;=\; \underbrace{R(\widehat{\theta}) - R_S(\widehat{\theta})}_{\mathcal{E}_{gen}} + \underbrace{R_S(\widehat{\theta}) - R_S(\bar{\theta})}_{\mathcal{E}_{opt}} + \underbrace{R_S(\bar{\theta}) - R_S(\theta_0)}_{\leq 0} + \underbrace{R_S(\theta_0) - R(\theta_0)}_{\mathbb{E}=0},$$

Since the last two parts are either negative or with zero expectation, we mainly focus on the first two parts, namely, generalization error $R(\widehat{\theta}) - R_S(\widehat{\theta})$ and optimization error $R_S(\widehat{\theta}) - R_S(\bar{\theta})$. Based on Hardt et al. (2016), $\mathcal{E}_{gen}$ is upper bounded by algorithmic stability.

**Uniform argument stability (UAS)**  UAS aims to quantify the output sensitivity in $\mathcal{L}_2$ norm w.r.t an arbitrary change in a single data point. An algorithm is $\lambda$-UAS if for neighboring datasets $S_1 \sim S_2$ (i.e., $S_1$ and $S_2$ differ only in a single data point), it satisfies that

$$\sup_{S_1 \sim S_2} \|\widehat{\theta}(S_1) - \widehat{\theta}(S_2)\| := \sup_{S_1 \sim S_2} \lambda(S_1, S_2) \leq \lambda.$$

where $\|\cdot\|$ represents the $\mathcal{L}_2$ norm. Under proper conditions, the UAS bound implies a generalization error bound (Bassily et al., 2020): if $P(\|\lambda(S_1, S_2)\| \geq \gamma) \leq \kappa_0$ for any neighboring $(S_1, S_2)$, then for any $\kappa$,

$$P\left[|\mathcal{E}_{gen}| \geq c\left(\gamma(\log n)(\log(n/\kappa)) + \sqrt{\frac{\log(1/\kappa)}{n}}\right)\right] \leq \kappa + \kappa_0. \qquad (3)$$

## 3.2 Upper and lower bound

This section presents the upper bound and lower bound of UAS of adversarial training when its natural counterpart is convex and smooth.

The upper bound of UAS of adversarial training can be directly extended from Bassily et al. (2020) as follows:

**Proposition 1.** *Assume $l(f_\theta(x), y)$ is L-Lipschitz and convex w.r.t. $\theta$, and $\theta \in B_2(0, r)$. The two models $\theta_1^{(t)}$ and $\theta_2^{(t)}$ are adversarial training estimators obtained using datasets $S_1, S_2$ respectively. For SGD,*

$$\sup_{S_1 \sim S_2} \mathbb{E}\left[\|\theta_1^{(T)} - \theta_2^{(T)}\|\right] = O\left(\min\left\{r, L\sqrt{\sum_{t=1}^{T} \eta_t^2} + L\frac{\sum_{t=1}^{T} \eta_t}{n}\right\}\right).$$

*The upper bound of GD is the same.*

The following theorem presents the lower bound of UAS. For simplicity, we consider the case of constant learning rate, i.e., $\eta_t = \eta$ for $t = 1, ..., T$.

**Theorem 1.** *Assume $\theta \in B_2(0, r)$. There exist some $\epsilon > 0$ and some loss function $l(f_\theta(x), y)$ which is differentiable and convex w.r.t. $\theta$, such that $\theta_1^{(t)}$ and $\theta_2^{(t)}$, which are SGD-based adversarial training estimators obtained using $S_1, S_2$ respectively under attack strength $\epsilon$, satisfies that*

$$\sup_{S_1 \sim S_2} \mathbb{E}\|\theta_1^{(T)} - \theta_2^{(T)}\| = \Omega\left(\min\left\{1, \frac{T}{n}\right\}\eta\sqrt{T} + \frac{\eta T}{n}\right).$$

*For GD, the lower bound is*

$$\sup_{S_1 \sim S_2} \|\theta_1^{(T)} - \theta_2^{(T)}\| = \Omega\left(\min\left\{1, \eta\sqrt{T} + \frac{\eta T}{n}\right\}\right).$$

To prove Theorem 1, similar to Bassily et al. (2020), we design a smooth clean loss function with two datasets $S_1 \sim S_2$ and study the exact change of the model parameters. The detailed proof is postponed to the Appendix D.

As discussed by Bassily et al. (2020), the non-smoothness of the loss is the main reason for poor stability. The presented low bounds match the result of Bassily et al. (2020), but it is worth mentioning that the two results are not directly comparable since Bassily et al. (2020) studied the UAS of clean training when the loss function $l(f_\theta(x), y)$ is non-smooth, while our work studies the UAS of adversarial training when the loss function is smooth. On the other hand, the UAS of cleaning training under smooth loss, implied by Theorem 3.8 of Hardt et al. (2016), is of order $O(\min\{r, L\sum_{t=t_0}^{T} \eta_t/n\})$. Therefore, we conclude that adversarial training has a worse stability than its natural counterpart.

To ensure the convergence of optimization (i.e., $\eta T$ is not so small) and the generalization performance (Proposition 1 and Theorem 1), one may take $T = n^2$ and $\eta = 1/n^{3/2}$. The resulting optimization error and stability then become $O(1/\sqrt{n})$, which matches the minimax lower bound of excess risk (Chen et al., 2018). However, such a choice of $(\eta, T)$ is impractical and needs to improve (refer to the discussion in Bassily et al., 2020).

## 3.3 The role of numerical attack error

In real-world applications, calculating the exact attack $A_\epsilon$ for general models is not easy, and usually, a numerical approximation $A_\epsilon'$ (e.g., by FGM or PGD) is used in the adversarial training algorithm.

Some recent literature start to aware the important impact of the numerical attack error (i.e., the difference between $A_\epsilon$ and $A'_\epsilon$). For example, Gao et al. (2019); Zhang et al. (2020) took account of the attack approximation method in the convergence analysis of adversarial training, and Deng et al. (2020) studied the convergence of PGD attack.

For algorithmic stability, extended from Proposition 1, the following result considers the effect of attack error. Comparing the upper bounds of Proposition 1 and Corollary 1, it suggests one to control the attack error carefully in the adversarial training.

**Corollary 1.** *Under the same conditions of Proposition 1, assume the algorithm uses an approximation $A'_\epsilon$ instead of the exact attack $A_\epsilon$ with attack error $\min \|A_\epsilon(x, y, w) - A'(x, y, w)\| \le \Delta\varepsilon$ for any $(x, y, w)$, where the minimum is taken when the exact attack (i.e., (1)) is not unique. Assume $\nabla_\theta l(f_\theta(x), y)$ is $\kappa$-Lipschitz w.r.t. $x$. Then, for SGD*

$$\sup_{S_1 \sim S_2} \mathbb{E}\|\theta_1^{(T)} - \theta_2^{(T)}\| = O\left(\min\left\{r, L\sqrt{\sum_{t=1}^{T}\eta_t^2} + L\frac{\sum_{t=1}^{T}\eta_t}{n} + \kappa\Delta\varepsilon\sum_{t=1}^{T}\eta_t\right\}\right).$$

*The upper bound of GD is the same.*

Besides the convex case, some discussions for non-convex case can be found in Appendix A. The observations are similar.

# 4 Improving the stability

In this section, we show that injecting noise in adversarial training enhances the smoothness of adversarial loss, and hence improves the stability of adversarial training.

## 4.1 Source of non-smoothness

As mentioned after Theorem 1, the non-smoothness issue in adversarial training is the main cause of the poor stability. Summarizing from the related works, we identify two important sources of non-smoothness in adversarial training even when the standard loss is smooth: (1) when the data are overfitted, i.e., the training loss is almost 0 and $\nabla_{x_i} l(f_\theta(x_i), y_i) \approx 0$, the adversary has no preference on the attack direction at $x_i$, and the numerical estimation of $A_\epsilon$ is not stable, which possibly leads to an unstable update iteration of adversarial training; (2) there exists a certain set of $\theta$, such that the adversarial training loss is always non-differentiable regardless of the training data, even when its natural counterpart is smooth. For example, in linear regression, when $\theta_t$ is closed to the null model, the non-smoothness issue occurs (Xing et al., 2021a).

To tackle both non-smooth issues, we propose incorporating noise injection in the training process as described in the following section.

## 4.2 Injecting noise during training

In this section, we present the noise injection algorithm in adversarial training and provide some theoretical justifications.

Algorithm 1 below illustrates the details of the noise injection method. The basic idea behind this is that: the non-smoothness of adversarial loss occurs only when $\theta$ and $x_i$' belong to a certain special region (e.g., in linear regression, when $\theta$ is closed to either zero or when $\theta^\top x_i \equiv y_i$), thus injecting some small noise to both $\theta$ and $x$ helps them to escape from such region where non-smoothness occurs, which further ensures the Lipschitz continuity property.

**Remark 1.** *The Gaussian noise in Algorithm 1 is for proof simplicity. In general, it can be changed to other noise distributions if the tail is not heavy.*

In the literature, there have been some applications of noise injection. For example, He et al. (2019) considered injecting noise to the weights as a regularization method to improve the adversarial robustness. Besides literature in supervised learning (Weng et al. (2018); Wang et al. (2018); Ford et al. (2019)), injecting noise in data was also considered to stabilize the training process of GAN (Arjovsky and Bottou, 2017; Jenni and Favaro, 2019).

**Algorithm 1** Add noise to weight and data

---

**Input:** data $\{(x_i, y_i)\}_{i=1}^n$, number of iterations $T$, learning rate $\{\eta_t\}_{t=1}^T$, attack strength $\epsilon$, noise size $(\xi_\theta, \xi_x)$, scale parameter $r$, initialization $\theta^{(0)}$.
**for** $t = 1$ **to** $T$ **do**
    Add Gaussian noise with variance $\xi_\theta^2$ to each dimension of $\theta_t$ to form $\widetilde{\theta}^{(t)}$, and add Gaussian noise with variance $\xi_x^2$ to each dimension of $x_t$ to obtain $\widetilde{x}_{i_t}$.
    Calculate the attack (based on $\widetilde{x}_{i_t}$ and $\widetilde{\theta}^{(t)}$) as $\widehat{z}_{i_t}$.
    Take gradient w.r.t $\widetilde{\theta}^{(t)}$ on $l(f_{\widetilde{\theta}^{(t)}}(\widehat{z}_{i_t}), y_{i_t})$.
    Update $\theta^{(t)}$ to $\theta^{(t+1)}$ with rate $\eta_t$.
    Project $\theta^{(t+1)}$ onto $B_2(0, r)$.
**end for**
**Output:** $\theta^{(T)}$.

---

In the following theorems, we provide a theoretical justification for the stability and optimization when injecting noise into adversarial training for the following models:

- Linear regression: $l(f_\theta(x), y) = (x^\top \theta - y)^2$.

- Logistic regression: $l(f_\theta(x), y) = -\log^{1\{y=1\}}(p) - \log^{1\{y=-1\}}(1 - p)$, where $p = p(x^\top \theta) = 1/(1 + e^{-x^\top \theta})$.

- Smooth hinge loss: the hinge loss $\max\{0, 1 - y(x^\top \theta)\}$ is not smooth at 0, hence is approximated by $l(f_\theta(x), y) = (1 - y(x^\top \theta))H((1 - y(x^\top \theta))/h)$, where $h > 0$ is a bandwidth parameter, and $H$ is a smooth approximation of the indicator function $I\{x \geq 0\}$. The detailed conditions on $H$ are postponed to Lemma 7 in the Appendix D.2.

The following assumption is imposed on the data:

**Assumption 1.** *The independent variable $x \in \mathbb{R}^d$ follows multivariate Gaussian distribution with zero-mean and $\Sigma$ whose eigenvalues are bounded and away from zero.*

*For regression, $\mathbb{E}|y|$ and $\mathbb{E}\|yx\|$ are finite. For some constant $C > 0$, any $\theta \in B_2(0, Cr)$ satisfies $P(|x^\top \theta - y| \in [\zeta_1 r, \zeta_2 r]) = O(\zeta_2 - \zeta_1)$ for $\zeta_1, \zeta_2 > 0$.*

*For classification, the label is $y \in \{\pm 1\}$. The upper bound $r$ satisfies $r/\max_{i=1,\dots,n} \|x_i\| \to 0$.*

The Gaussian assumption in $x$ is merely for derivation simplicity. The assumptions w.r.t. regression avoids $|x^\top \theta - y|$ from clustering around zero when $\|\theta\|/r$ approaches zero. A linear model $\mathbb{E}[y|x] = \theta_0^\top x$ with Gaussian noise satisfies Assumption 1.

Given the above problems and data generating models, the following lemma studies the smoothness (i.e., the Lipschitz constant) of $\nabla_\theta l(f_\theta(x), y)$, and of the gradient of noise injected adversarial loss.

**Lemma 1** (Informal Statement for Lemma 3)**.** *Assume Assumption 1 holds. Denote $L$ as the Lipschitz constant of $l(f_\theta(x), y_j)$ w.r.t. $\theta$ for any $x \in B_2(x_j, 2\epsilon)$ and all $1 \leq j \leq n$. Then, in probability, $L$ is bounded by some finite $L^*$. Take the noise injected in data as zero-mean Gaussian with variance $(\xi_0^2/d)I_d$, and the noise injected in parameters is zero-mean Gaussian with variance $(\xi^2/d)I_d$ where $\xi = \xi_0 L^*$. Denote $E(\theta + \delta, \widetilde{x}, y)$ as the event that $\nabla_\theta l(f_{\theta+\delta}(\widetilde{x} + A_\epsilon(f_{\theta+\delta}, \widetilde{x}, y)), y)$ is $B^*/\zeta$-Lipschitz for some $B^* > 0$. There exists some choice of $(\xi, \zeta) \to 0$ such that with probability tending to one over the generation of $S$, uniformly for all $\theta \in B_2(0, r)$,*

$$P(E^c(\theta + \delta, \widetilde{x}, y)|(x, y) \in S) = o(1).$$

*The formal statement is postponed to Lemma 3 in the appendix.*

Let $P(E^c|S) := \sup_{\theta \in B_2(0,r),(x,y)} P(E^c(\theta + \delta, \widetilde{x}, y)|(x, y) \in S)$ in what follows, for notation simplicity.

**Remark 2.** *The terms $r$, $L$ are generic representations. For different loss functions and data dimension $d$, their values may change. In addition, the exact rate of $P(E^c|S)$ is affected by the value of $r, L$ as well as $\xi, \zeta$. We postpone the details to Appendix D.2 during the proof.*

The following lemma is an intermediate step in the derivation of Theorem 2 below, and reveals the important role played by $B^*/\zeta$. Note that since Lemma 1 holds over the randomness of $S$, instead of **uniform** argument stability, we turn to a bound similar to **hypothesis** stability (Bousquet and Elisseeff, 2002) for the following results. To simplify the representation, the values of $(B^*, L^*, r)$ are treated as constants in the following main text.

**Lemma 2.** *Under the same conditions as in Lemma 1, uniformly for all $i = 1, ..., n$, with probability tending to one over the generation of $S_1 \sim S_2$ where the $i$-th sample is replaced, for both GD and SGD, given $\|\theta_1^{(t-1)} - \theta_2^{(t-1)}\| = \Delta_{t-1}$, it follows that*

$$\mathbb{E}[\|\theta_1^{(t)} - \theta_2^{(t)}\|^2 | S_1, S_2, \Delta_{t-1}] \leq \left(1 + 2\eta_t^2 \frac{(B^*)^2}{\zeta^2} 1\{\eta_t \geq \frac{\zeta}{B^*}\}\right) \Delta_{t-1}^2 + reminder,$$

*where the detail of reminder term can be found in (8) in Appendix D.2. Note that the expectation taken on $\|\theta_1^{(t)} - \theta_1^{(t)}\|^2$ in GD is over the injected random noise, and the one for SGD is taken for both the sampling in SGD and the injected random noise.*

Lemma 2 illustrates the relationship between $\|\theta_1^{(t)} - \theta_2^{(t)}\|^2$ and $\Delta_{t-1}^2$. Recall that $B^*/\zeta$ is the Lipschitz constant of $\nabla_\theta l(f_{\theta+\delta}(\tilde{x} + A), y)$. When $\eta_t \geq \zeta/B^*$, a larger Lipschitz constant implies a larger upper bound of stability. When taking $\eta_t < \zeta/B^*$, we have the following result:

**Theorem 2.** *Under the same conditions as in Lemma 1, when taking $\eta_t \leq \zeta/B^*$, for both GD and SGD, with probability tending to one (where the probability refers to the generation measure of the $n + 1$ distinct independent samples in $S_1 \sim S_2$),*

$$\mathbb{E}[\|\theta_1^{(T)} - \theta_2^{(T)}\| | S_1, S_2] = O\left(\left[\sqrt{P(E^c|S_1) + P(E^c|S_2)} + \sqrt{\frac{1}{n}}\right]\sqrt{\sum_{t=t_0}^{T} \eta_t^2}\right)$$
$$+ O\left(\left[\Delta\varepsilon + \frac{1}{n} + P(E^c|S_1) + P(E^c|S_2)\right]\sum_{t=t_0}^{T} \eta_t\right).$$

Furthermore, extending from Lemma 9 of Bousquet and Elisseeff (2002), the generalization gap is upper bounded using hypothesis stability as follows.

**Proposition 2.** *Assume $\theta \in B_2(0, r)$. Denote $\widehat{\theta}(S)$ as the model obtained based on dataset $S$ using some algorithm. Assume $l(f_\theta(x), y) \in [0, M]$ when $\|x\| \leq \sqrt{d \log n}$, we have for any $i = 1, ..., n$,*

$$\mathbb{E}\left[\left(R(\widehat{\theta}(S_1)) - R_{S_1}(\widehat{\theta}(S_1))\right)^2\right] \leq \frac{M^2}{2n} + 4\mathbb{E}\left[\sup_{\theta \in B_2(0,r)} l^2(f_\theta(x + A_\epsilon), y)1\{\|x\| \geq \sqrt{d \log n}\}\right]$$

$$\tag{4}$$

$$+ 3M\mathbb{E}\left[\left|l(f_{\widehat{\theta}(S_1)}(x_i + A_\epsilon), y_i) - l(f_{\widehat{\theta}(S_2^i)}(x_i + A_\epsilon), y_i)\right|\right],$$

*where $S_2^i$ represents the neighboring dataset of $S_1$ whose $i$th sample is replaced. The notion $A_\epsilon$ is an abbreviation of the attack $A_\epsilon(f, x, y)$ or $A_\epsilon(f, x_i, y_i)$.*

Note that the last term on the RHS of (4) can be bounded according to the result of Theorem 2, under Lipschitz condition of the adversarial loss $l(f_\theta(x_i + A_\epsilon), y_j)$. The second term on the RHS of (4) can be bounded given some further conditions on the tail behavior of loss function.

Compared with the generalization upper bound obtained in Bousquet and Elisseeff (2002), in Proposition 2, there is an extra term corresponding to $\|x\| > \sqrt{d \log n}$. In Proposition 2, we only assume $l(f_\theta(x), y) \in [0, M]$ when $\|x\| \leq \sqrt{d \log n}$, which is weaker than the uniform bounded assumption in Bousquet and Elisseeff (2002).

Besides, we also establish the optimization error bound. The following theorem presents the convergence of noise-injected adversarial training when $\eta_t \equiv \eta$. For the proof, one can refer to the Appendix D.2.

**Theorem 3.** *Under the same conditions as in Lemma 1, for both GD and SGD, $\bar{\theta} :=$ $\mathrm{argmin}_{\theta \in B_2(0,r)} R_S(\theta)$, when $\eta_t \equiv \eta$,*

$$\mathbb{E}\left[\min_{t=1,\ldots,T} R_S(\theta^{(t)}) - \min_{\theta \in B_2(0,r)} R_S(\theta)\Big| S\right] \leq \frac{\mathbb{E}\|\theta^{(0)} - \bar{\theta}\|^2}{2\eta T} - \frac{\mathbb{E}[\|\theta^{(T)} - \bar{\theta}\|^2|S]}{2\eta T}$$
$$+ \frac{\eta(L^*)^2}{2} + O(L^*\xi) + O(r\Delta\varepsilon).$$

Theorem 3 presents the convergence of adversarial training loss throughout the training. Under the boundedness of $\theta$ and $\xi = \xi_0 L^*$, when $(\eta, T, \xi_0)$ is chosen properly (e.g. $(\eta T) \to \infty$ and $\xi_0 \to 0$) and $\Delta\epsilon \to 0$, the adversarial training loss converges to its minimal asymptotically.

It is noteworthy that the general design of Algorithm 1 does not specify the noise distribution.

While in Section 4.3, we use Gaussian noise to justify our theorems under linear regression empirically, different forms of noise can be utilized for complex models, refer to our experiments on deep neural networks in Section C.2.

**Remark 3.** *If an intercept term exists in the loss function, e.g., $l = (x^\top \theta + b - y)^2$ for linear regression, the analysis is similar to Lemma 1, leading to the same final conclusions as in Theorem 2 and 3.*

**Remark 4** ($\mathcal{L}_\infty$ Attack in Adversarial Training). *In general, the stability of $\mathcal{L}_\infty$ adversarial training is worse. To set an example, we consider the linear regression setup. For $\mathcal{L}_2$ attack, the gradient of adversarial loss is not Lipschitz only when $\theta$ approaches zero or $\theta^\top x$ is closed to $y$. Under $\mathcal{L}_\infty$ attack, the adversarial loss becomes*

$$(x^\top \theta - y)^2 + \epsilon^2\|\theta\|_1^2 + 2\epsilon\|\theta\|_1|x^\top \theta - y|,$$

*indicating there is a much larger set where the $\mathcal{L}_\infty$ adversarial loss is not smooth.*

*Noise injection is still helpful to remedy the non-smooth issue for $\mathcal{L}_\infty$ adversarial training and leads to results similar to Theorems 2 and 3. However, one will derive a worse upper bound for the stability and optimization error. Refer to Appendix E for more detailed arguments.*

### 4.3 Numerical illustration

We use simulation to illustrate how noise-injected adversarial training affects performance. In short, the quality of the updating gradient is better after injecting noise. The Lipschitz constant of $\nabla_\theta l(f_\theta(x + A), y)$ in Lemma 2 is smaller.

We consider linear regression problem in this experiment. The data is generated using $y = x^\top \theta^* + \delta$ with $x \sim N(0, I_d)$ with $d = 10$ and $\delta \sim N(0, \sigma^2)$. The coefficient $\theta_0$ is taken as $\theta_i^* = 1/\sqrt{d}$ for $i = 1, \ldots, d$. The variance of noise is taken as $\sigma^2 = 4$ and attack strength is $\epsilon = 2$. We randomly generated $n = 1000$ samples.

To train the regression model, we train $T = 500$ epochs with learning rate $\eta = 0.01$ and initialization $\theta^{(0)} = \mathbf{0}$. In each iteration, we calculate $\|\nabla f_t(\theta^{(t)}) - \nabla f_{t-1}(\theta^{(t-1)})\|/\|\theta^{(t)} - \theta^{(t-1)}\|$ as an approximation for the Lipschitz constant of the gradient, where $\nabla f_t(\theta)$ is the averaged gradient of adversairal loss for the $t$th batch of data $S_t$. Based on Lemma 2, a larger Lipschitz constant ($B^*/\zeta$) indicates a worse stability, which is the right tail of the histogram. The results are summarized the histograms in Figure 1.

From the left three histograms in Figure 1, one can see that injecting noise on parameters and data (where we set $\xi_x = \xi_\theta = \xi$) leads to a smaller distribution of Lipschitz constant for $\nabla f_t(\theta^{(t)})$, in terms of right tail percentile. For the right two histograms in Figure 1, a smaller batch size implies a heavier tail in the distribution of Lipschitz constant due to larger stochastic noise in estimating $\nabla f_t(\theta^{(t)})$.

Besides the experiment showing how noise injection affects the training process, we also conduct a simulation to illustrate the effect of attack error on the generalization. Due to the space limit, the simulation is postponed to the Appendix B. To briefly summarize the observations, for all scenarios we consider, when there is an error when calculating the attack, the generalization gap becomes larger.

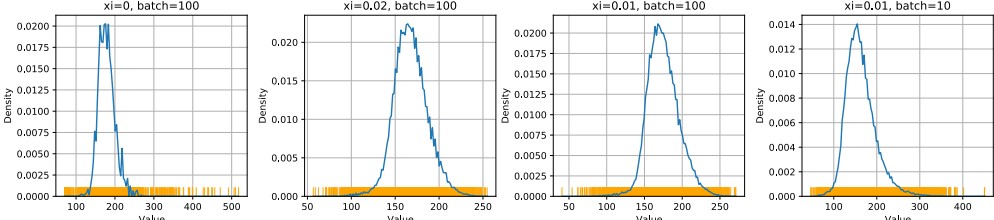

Figure 1: Density of $\|\nabla f_t(\theta^{(t)}) - \nabla f_{t-1}(\theta^{(t-1)})\|/\|\theta^{(t)} - \theta^{(t-1)}\|$. $\theta_{0,i} = 1/\sqrt{d}$ for $i = 1, ..., d$ with $d = 10$. $n = 1000$, $\sigma = 2$, $\epsilon = 2$. $\eta = 0.01$, $T = 500$. Vanishing initialization. A larger $\xi$ implies a smaller Lipschitz gradient of $\nabla f_t(\theta^{(t)})$. The (mean, sd, 99.9%-quantile) are (178.90, 22.34, 323.36), (167.72, 19.20, 234.43), (174.95, 20.71, 247.93), (163.05, 34.89, 337.07) for the above four histograms.

## 5 Exploration in Neural Networks

While our main contributions are for statistical models, we also provide some theoretical results and numerical experiments associated with neural networks.

It is still an open question how to connect existing algorithmic stability tools to neural networks. Since the number of parameters in neural networks is much larger than simple models, a simple bound on $\|\theta_1^{(T)} - \theta_2^{(T)}\|$ is not useful. Instead, we consider a two-layer neural network with lazy training and vanishing initialization in regression and provide a stability bound directly on the loss. In order to track the neural network parameters, we track both the convergence and the stability together. This is more restrictive than simple models.

The following (informal) statement presents the stability of two-layer nonlinear networks with lazying training in adversarial training setup. The formal statement of the theorem is postponed to Appendix C.1. Based on the following theorem, under proper configurations, the noise-injected training in neural networks improve the stability:

**Theorem 4** (Informal Statement). *For two-layer nonlinear (including ReLU) networks, with proper initialization and training configurations, training only on the hidden layer with proper noise injection, it satisfies that*

$$\mathbb{E}_{S_1 \sim S_2} \left| l(f_{\theta_1^{(T)}}[x + A_\epsilon(f_{\theta_1^{(T)}}, x_i, y_i)], y_i) - l(f_{\theta_2^{(T)}}[x + A_\epsilon(f_{\theta_2^{(T)}}, x_i, y_i)], y_i) \right|$$

$$= O\left( \left[ L\sqrt{P(E^c)} + \sqrt{\frac{L^2}{n}} \right] \eta\sqrt{T} + \left[ \frac{L}{n} + LP(E^c) \right] \eta T \right) + rem,$$

*where $rem = o(1)$ and is not the dominant term.*

There are two differences between Theorem 4 and the results in simple models. First, it is not useful to directly assume the weights of the neural network parameters within a large ball and put this large number into the bound, thus we simultaneously study the convergence and stability of the neural network to tighten the stability bound. Second, instead of bounding the stability of the parameters, we turn to bound the stability of the loss, which is more meaningful to this over-parameterized method.

Besides the results in two-layer networks, we also numerically study the generalization gap using deep neural networks with CIFAR10 dataset. Due to space limit, we postpone the experiments to Appendix C.2. The observations from numerical experiments are (1) injecting noise can reduce the generalization gap between training and testing performance, and (2) improving the accuracy of attack also improves the quality of the adversarial training. Both of the observations are similar to those in simulations.

## 6 Conclusion

In this paper, we evaluate the algorithmic stability of the adversarial training method. Based on the lower bound and upper bound of UAS, we reveal that the naive adversarial training is not as stable

as its natural counterpart. To improve the stability, we argue that it is helpful to inject noise into model parameters and input data. Our theory verifies the effectiveness of noise injection in some simple models. Besides, our theory also considers the effect of attack error and indicates that the upper bound of UAS is smaller when the attack error is smaller.

The above theoretical investigations emphasize the usage of noise injection and controlling numerical attack error during the adversarial training. These theoretical insights are well validated by our simulations under simple regression models.

There are two future research directions motivated by this study. Although we observe a similar phenomenon in deep neural networks as our theory in simple models, there is a gap between the exact algorithmic stability of deep neural networks and the UAS bounds in simple models. Our analysis in the two-layer neural networks is a trail in this area, but a more comprehensive study study on algorithmic stability of the deep neural network is wanted. Second, as we mentioned in the numerical experiments, a wider neural network has a poor attack. Corresponding theoretical explanation is also an interesting topic.

# 7 Acknowledgements

This project is partially supported by NSF-SCALE MoDL(2134209) and NSF-DMS-1811812.

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
