## A    General loss

The following theorem provides a more general upper bound for the algorithm stability beyond convex $l(f_\theta(x, y))$.

**Theorem 5.** *Assume $\|\theta\|$, $\|x\|$ are both bounded, and we choose $\eta_t \leq c/t$ for some $c > 0$. If , If $l(f_\theta(x, y))$ is L-Lipschitz w.r.t. $\theta$, $(\nabla_\theta l(f_\theta(x), y), \nabla_x l(f_\theta(x), y))$ is $\kappa$-Lipschitz in $(\theta, x)$, and the attack error is smaller than $\Delta\varepsilon$, then the UAS upper bound for SGD becomes*

$$\sup_{S_1 \sim S_2} \mathbb{E}\|\theta_1^{(T)} - \theta_2^{(T)}\| = O\left((\kappa\Delta\varepsilon + L)\left(\frac{T}{n}\right)^{\frac{c\kappa}{c\kappa+1}}\right),$$

*and the one for GD is*

$$\sup_{S_1 \sim S_2} \|\theta_1^{(T)} - \theta_2^{(T)}\| = O\left((\kappa\Delta\varepsilon + L)\, T^{c\kappa}\right).$$

By Theorem 5, to obtain better stability, it suffices to control the number of iterations and the attack error. To ensure a diminishing UAS upper bound, the choice of $T$ should be much smaller than $n$. Note that SGD only has a small probability of encountering the exact different data points between the two datasets at the very first iteration, leading to a smaller UAS upper bound than GD due to the diminishing learning rate.

**Remark 5.** *Compared with the UAS bound $O(T^{\frac{c\kappa}{c\kappa+1}}/n)$ for standard training (Hardt et al., 2016), Theorem 5 also suggests that adversarial training prefers a smaller number of steps to reduce the corresponding UAS upper bound. This echos the observations in Rice et al. (2020) that early stopping is necessary in adversarial training.*

**Remark 6.** *The class of functions considered in Proposition 1 (smooth convex function) is not a special case of those in Theorem 5, so results of Proposition 1 and Theorem 5 are not directly comparable.*

## B    Additional Experimental Results

Below is the simulation study mentioned in Section 4.3 on the effect of attack error.

In this experiment, we follow the data generating model as for Figure 1 and take $\epsilon = 0.2$. Unlike noise injection which changes the Lipschitz constant $B^*/\zeta$, from Lemma 2 and Theorem 2, attack error has little effect on the Lipschitz constant, and directly deteriorates the stability. Consequently, instead of showing the Lipschitz constant (Figure 1), we present the generalization error.

To create error in the attack, for each data, after we obtain the true attack using analytical solution, denoting as $\widehat{z}_i$, we create random noise $\delta_{z_i} \sim N(0, (\sigma_\epsilon^2/d)I_d)$, and finally project $\delta_{z_i} + \widehat{z}_i$ onto $B_2(x_i, \epsilon)$ to obtain the corrupted attack. There is no noise injection for parameters and $x_i$ in this experiment, i.e. $\delta_\xi = \delta_x = 0$.

We take $(\epsilon, \delta_\epsilon)$ as $(0.2, 0)$, $(0.2, 0.1)$, $(0.4, 0)$, $(0.4, 0.2)$. For each setup, the experiment repeats 100 times to obtain a mean and variance. The results are summarized in Figure 2 for the whole training process and Table 1 for the detailed values at the 500th iteration. Compared with $\delta_\epsilon = 0$, when $\delta_\epsilon > 0$, both adversarial testing loss and generalization error increases. For the increase in adversarial training loss, our conjecture is that the attack error perturbs the training process, so the trained model is slight away from its optimum.

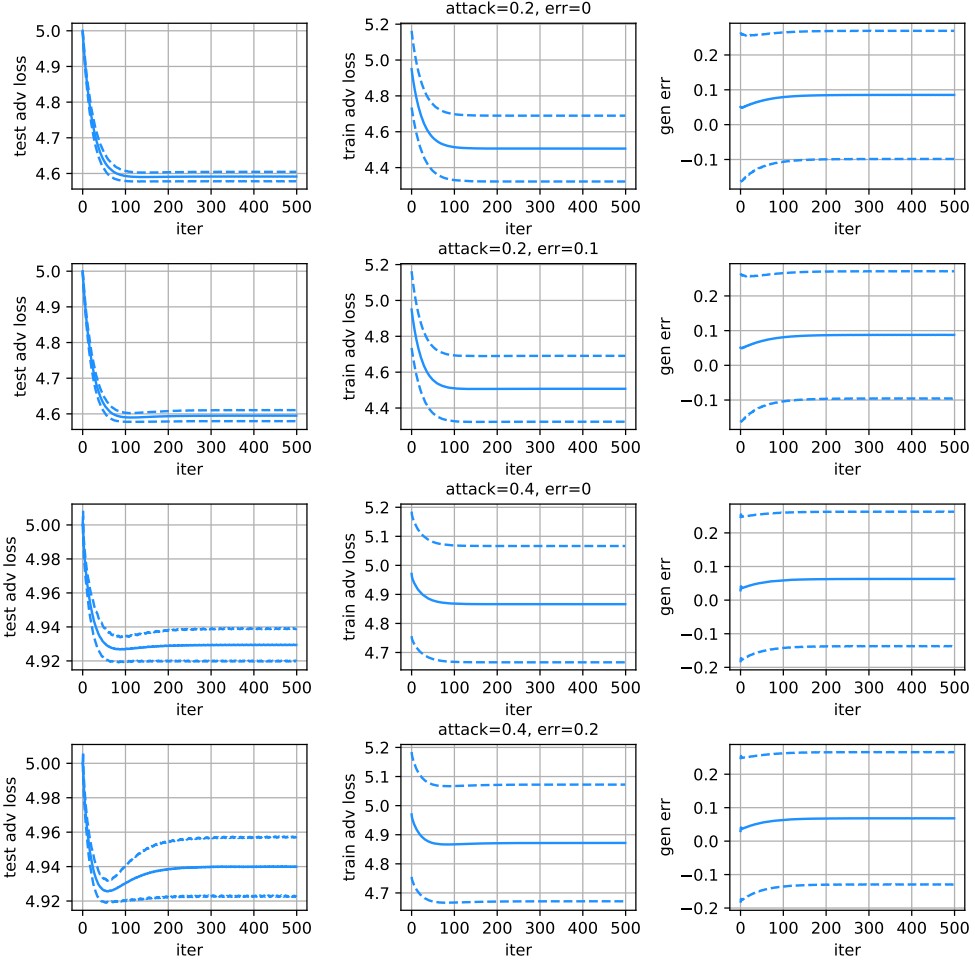

Figure 2: Mean and standard deviation of adversarial testing loss (left), adversarial training loss (middle), generalization error (right) for four groups of $(\epsilon, \sigma_\epsilon)$. After injecting error into attack (2nd and 4th row), all the adversarial testing loss, adversarial training loss, and generalization error increase (compared with 1st and 3rd row).

| $\epsilon$ | $\sigma_\epsilon$ | Test adv loss | Train adv loss | Gen err |
|---|---|---|---|---|
| 0.2 | 0.0 | 4.5914 (0.0134) | 4.5061 (0.1836) | 0.0854 (0.1838) |
| 0.2 | 0.1 | 4.5950 (0.0157) | 4.5073 (0.1837) | 0.0877 (0.1833) |
| 0.4 | 0.0 | 4.9294 (0.0096) | 4.8664 (0.2003) | 0.0631 (0.1998) |
| 0.4 | 0.2 | 4.9399 (0.0175) | 4.8718 (0.2005) | 0.0681 (0.1977) |

Table 1: Mean and standard deviation of adversarial testing loss, adversarial training loss, generalization error for the 500th iteration. After injecting error into attack, the adversarial testing loss, adversarial training loss, and generalization error increase.

## C Results in Neural Networks

### C.1 Two-Layer Neural Network

This section provides some results w.r.t two-layer neural network with lazy training, i.e. training the first hidden layer, and vanishing initialization.

Denote $h$ as the number of hidden nodes, $\theta = [\theta_1 \mid \theta_2 \mid ... \mid \theta_h]$ are weights for each node. The neural network outputs value as follows

$$f(x, \theta) = \frac{1}{\sqrt{h}} \sum_{j=1}^{h} a_j \phi(\theta_j^\top x),$$

where $\phi$ is a smooth activation function which is either (1) twice differentiable and $\phi(0) = 0$, or (2) ReLU activation function. The value of $a_j$'s are unchanged after the initialization, and we randomly generate them from $\{\pm 1\}$. The initial value of $\theta_j$ ($\theta_j^{(0)}$) is generated from $N(0, I_d/(dh^{\delta+1}))$, i.e. vanishing initialization. The adversarial training will update $\theta_j^{(t)}$ throughout training.

We consider regression task, and to simplify the analysis, we use the following data generation model:

$$y = \theta_0^\top x + \omega,$$

where $x$ follows $N(0, I_d)$, $\|\theta_0\|$ is a constant, and $\omega$ is a Gaussian noise with zero mean and finite variance.

Also denote adversarial risk minimizor and the minimal adversarial risk for linear model as follows:

$$\theta^* = \operatorname*{argmin}_\theta \mathbb{E}(y - \theta^\top(x + A_\epsilon(f_\theta, x, y)))^2, \; R^* = \min_\theta \mathbb{E}(y - \theta^\top(x + A_\epsilon(f_\theta, x, y)))^2.$$

We have the following theorem:

**Theorem 6.** *Assume $\Delta\epsilon = 0$. Under the model setup of neural network and data generating model in this section, assume $\log n \sqrt{d^2/n} \to 0$, $(d \log n)/\sqrt{h} \to 0$ and $\sqrt{d \log n}(1 + D)^T/h^{\delta/2} \to 0$ for some large constant $D$. Take $L = \Theta(\sqrt{d \log n})$. If $\eta = \zeta/B^*$, $T = (\log \log n)/\eta$, and $\sqrt{d \log n} \log(hT)\xi \to 0$, then using GD, taking $(x, y)$ as a random testing data,*

$$\mathbb{E}_{S_1 \sim S_2} \left| l(f_{\theta_1^{(T)}}[x + A_\epsilon(f_{\theta_1^{(T)}}, x_i, y_i)], y_i) - l(f_{\theta_2^{(T)}}[x + A_\epsilon(f_{\theta_2^{(T)}}, x_i, y_i)], y_i) \right|$$

$$= O\left(\left[L\sqrt{P(E^c)} + \sqrt{\frac{L^2}{n}}\right] \eta\sqrt{T} + \left[\frac{L}{n} + LP(E^c)\right] \eta T\right) + rem, \tag{5}$$

*where $rem = o(1)$ and is not the dominant term when $h$ and $\delta$ are large enough. The value $B^*$ is defined similarly as in Lemma 5 for linear regression.*

Theorem 6 illustrates how the hypothesis stability changes throughout the training. This stability result can be trivially used to bound the third term on the RHS of generalization inequality 4 in Proposition 2.

To prove Theorem 6, it is harder than simple models since $\|\theta_1^{(T)} - \theta_2^{(T)}\|$ is not so meaningful in neural networks. Starting from the vanishing initialization, we track the change in each node one by one. We only consider GD because GD ensures the convergence of neural network.

For the two terms in (5), the first term is obtained when bounding the difference between $\theta_1^{(T)}$ and $\theta_2^{(T)}$; the second term counts for (1) the rare events on $S_1, S_2$ (we mention "with probability tending to 1 over $S_1 \sim S_2$" in Lemma 2) ;(2) the difference when we use linear network to approximate nonlinear network.

*Proof.* We provide the proof for the first term in (5) for smooth activation function. The idea is similar to the arguments in Xing et al. (2021a). We first consider how linear network with zero initialization could act, compared to a linear model with zero initialization under noise injection.

Then we bound the difference between noise-injected linear network with zero initialization and noise-injected nonlinear network with vanishing initialization.

First, we consider a linear network

$$f_L(x, \theta) = \frac{1}{\sqrt{h}} \sum_{j=1}^{h} \phi'(0) a_j \theta_j^\top x = \left( \frac{1}{\sqrt{h}} \sum_{j=1}^{h} \phi'(0) a_j \theta_j \right)^\top x.$$

When injecting noise $\delta_j$ into $\theta_j$, it becomes

$$f_L(x, \theta + \delta) = \frac{1}{\sqrt{h}} \sum_{j=1}^{h} \phi'(0) a_j (\theta_j + \delta_j)^\top x = \left( \frac{1}{\sqrt{h}} \sum_{j=1}^{h} \phi'(0) a_j \theta_j \right)^\top x + \left( \frac{1}{\sqrt{h}} \sum_{j=1}^{h} \phi'(0) a_j \delta_j \right)^\top x,$$

where $\frac{1}{\sqrt{h}} \sum_{j=1}^{k} \phi'(0) a_j \delta_j$ is a random vector with zero mean and covariance $N(0, \phi'(0)^2 \xi^2 I_d/d)$.

Denote the parameters trained from linear model with zero initialization as follows: for dataset $S_k$,

$$\theta_{j,k}^{OP1}(0) = \mathbf{0},$$

$$\theta_{j,k}^{OP1}(t+1) = \theta_{j,k}^{OP1}(t) - \eta \left( \frac{2}{n} \sum_{(x,y) \in S_k} \frac{a_j \phi'(0)}{\sqrt{h}} (x + A_\epsilon(f, x, y)) \left( \frac{\phi'(0)}{\sqrt{h}} \sum_{m=1}^{h} a_j (\theta_{m,k}^{OP1}(t) + \delta_m)^\top x - y \right) \right).$$

As a result, injecting noise $\delta_j \sim N(0, \xi^2 I_d/d)$ into a linear network is equivalent to injecting noise $N(0, \phi'(0)^2 \xi^2 I_d/d)$ into a linear model. So one may use arguments in Lemma 2 and Theorem 2 to study

$$\left\| \frac{1}{\sqrt{h}} \sum_{j=1}^{h} \phi'(0) a_j (\theta_{j,1}^{OP1}(t) - \theta_{j,2}^{OP1}(t)) \right\|$$

given $\left\| \frac{1}{\sqrt{h}} \sum_{j=1}^{h} \phi'(0) a_j (\theta_{j,1}^{OP1}(t-1) - \theta_{j,2}^{OP1}(t-1)) \right\|$.

On the other hand, from the updating rule of $\theta_{j,k}^{OP1}(t)$, one can also see that $\theta_{j,k}^{OP1}(t) \equiv \theta_{l,k}^{OP1}(t)$ if $a_j = a_l$. Therefore, besides $\frac{1}{\sqrt{h}} \sum_{j=1}^{h} \phi'(0) a_j \theta_{j,k}^{OP1}(T) \to \theta^*$ (since we are using GD and $n \to \infty$), one can also solve $\theta_{j,k}^{OP1}(T)$ for each $j$.

Denote the parameters trained from nonlinear model with vanishing initialization as follows: for dataset $S_k$,

$$\theta_{j,k}^{OP2}(t+1)$$
$$= \theta_{j,k}^{OP2}(t) - \eta \left( \frac{2}{n} \sum_{(x,y) \in S_k} \frac{a_j \phi'((\theta_{m,k}^{OP2}(t) + \delta_m)^\top x)}{\sqrt{h}} (x + A_\epsilon(f, x, y)) \left( \frac{1}{\sqrt{h}} \sum_{m=1}^{h} a_j \phi((\theta_{m,k}^{OP2}(t) + \delta_m)^\top x) - y \right) \right).$$

When we do not inject noise to network parameters, the difference between $\theta_{j,k}^{OP2}(t)$ and $\theta_{j,k}^{OP1}(t)$ can be neglected when $(d \log n)/\sqrt{h} \to 0$ and $\sqrt{d \log n}(1 + D)^T / h^{\delta/2} \to 0$ (Theorem 3 of Xing et al. (2021a)).

When injecting noise to network parameters, we further want $\max_{i,k} |\delta_k^\top x_i| \to 0$ in probability so that $\phi((\theta_j + \delta_j)^\top x) = \phi(0) + \phi'(0)(\theta_j + \delta_j)^\top x + O(((\theta_j + \delta_j)^\top x)^2)$ and the second order term is a remainder. Note that since $\delta_k$ and $x_i$ are generated from Gaussian, we have $\|\delta_k\|/\xi$ is in $O(\sqrt{\log(hT)})$ (among the $T$ iterations and $h$ nodes in each iteration) and $\|x_i\|$ is in $O(\sqrt{d \log n})$. As a result, $\max_{i,k} |\delta_k^\top x_i| = O(\xi \sqrt{d \log n \log(hT)})$, which by assumption is a vanishing term. Further injecting noise in data has little impact on the difference between $\theta_{j,k}^{OP2}(t)$ and $\theta_{j,k}^{OP1}(t)$ since $\xi_0 \to 0$, and we skip this part.

The proofs for ReLU networks follows similar arguments with assistance of Theorem 4 of Xing et al. (2021a).

$\square$

## C.2 Exploration in deep learning

Section 4.2 and 3.3 suggest that (1) the noise injection in model parameters and input data, and (2) better accuracy on the approximation of attack $A_\epsilon$, both lead to a better stability of the algorithm. Although these theoretical insights are derived under certain simple statistical models, we conjecture that they apply to modern complex models. Hence, in this section, we assess their effects on the stability of DNNs. Inspired by Proposition 3, we use the difference between adversarial training accuracy and adversarial testing accuracy to measure algorithm stability.

We use SGD with batch size 128 and weight decay 0.0002 as the optimizer. The learning rate is taken as 0.1 at the beginning and multiplies 0.1 at the 75th and 90th epoch. The total number of epochs is 100. To overcome the non-smoothness from ReLU activation, if not specified, we use WideResNet34-1 as the network structure with replacing backward update of ReLU into Softmax(10) using the `BPDA` in AdverTorch[2] (Ding et al. 2019).

### C.2.1 Noise injection

Theorem 2 indicates that injecting noise in adversarial training improves algorithmic stability. This section examines the effect of noise injection in adversarial training under a deep learning setup.

In this experiment, we use CIFAR10 and compare the adversarial testing accuracy before/after injecting noise using the implementation of TRADES[3] in Zhang et al. (2019). For the model parameters, the noise for each element is generated from a zero-mean normal distribution with a standard deviation equal to $\alpha_{h,t}\sigma_{h,t}$, where $\sigma_{h,t}^2$ is the variance of the parameters in $h$-th layer at $t$-th iteration and $\alpha_{h,t}$ is a trainable parameter initialized as 0.1. The implementation follows the one in He et al. (2019)[4]. We consider data augmentation (Shorten and Khoshgoftaar, 2019) as a form of noise injection to the data, and compare generalization performance with/without data augmentation during the training. For data augmentation method, we follow Zhang et al. (2019); Wang et al. (2019b); He et al. (2019) to include `Randomop(32, padding=4)` and `RandomHorizontalFlip()`. Each setup is repeated for five time to obtain a mean and variance. The results are summarized in Table 2. AT represents the vanilla adversarial training.

| # | Method | Noise | Aug | $\epsilon$ | Adv Train Acc | Adv Test Acc | Gen Gap | Std(Gen Gap) |
|---|--------|-------|-----|-----|---------------|--------------|---------|--------------|
| 1 | AT | No | No | $\mathcal{L}_2$ 0.5 | 97.1775 | 55.4525 | 41.725 | 0.3414 |
| 2 | AT | No | Yes | $\mathcal{L}_2$ 0.5 | 72.542 | 52.698 | 19.844 | 1.302 |
| 3 | AT | Yes | Yes | $\mathcal{L}_2$ 0.5 | 69.27 | 54.72 | 14.56 | 1.412 |
| 4 | AT | No | No | $\mathcal{L}_\infty$ 8/255 | 75.525 | 37.055 | 38.47 | 0.2493 |
| 5 | AT | No | Yes | $\mathcal{L}_\infty$ 8/255 | 50.486 | 36.162 | 14.324 | 0.5118 |
| 6 | AT | Yes | Yes | $\mathcal{L}_\infty$ 8/255 | 47.94 | 37.64 | 10.30 | 1.252 |
| 7 | TRADES | No | No | $\mathcal{L}_2$ 0.5 | 91.42 | 51.555 | 39.865 | 0.781 |
| 8 | TRADES | No | Yes | $\mathcal{L}_2$ 0.5 | 67.632 | 57.0 | 10.632 | 0.7764 |
| 9 | TRADES | Yes | Yes | $\mathcal{L}_2$ 0.5 | 65.904 | 61.508 | 4.396 | 0.4583 |

Table 2: Effect of different methods on generalization gap under $\mathcal{L}_2/\mathcal{L}_\infty$ attack.

In Table 2, when training using adversarial training for (#1, #2, #3), both data augmentation and noise injection in model parameters reduce the accuracy difference a lot. The vanilla adversarial training/testing gap is around 42%. After introducing data augmentation, the gap reduces to 20%. Finally, after further injecting noise into model parameters, the gap gets sown to only 15%. We also compare the adversarial testing accuracy before/after injecting noise for $\mathcal{L}_\infty$ attack. The observations are similar. The observation from TRADES are similar to AT. It still suffers from the stability issue, and injecting noise can reduce the generalization gap.

It is worth emphasizing that the aim of injection noise is not to improve the final testing performance, but to reduce the generalization gap (without sacrificing the testing performance). It has been shown in Rice et al. (2020) that, the adversarial training tends to overfit, i.e., the high training

---

[2] `https://github.com/BorealisAI/advertorch`
[3] `https://github.com/yaodongyu/TRADES`
[4] `https://github.com/elliothe/CVPR_2019_PNI/blob/master/code/models/noise_layer.py`

acc is deceptive and not reliable. Therefore, the training loss shall decrease if a stable algorithm is implemented.

**Remark 7.** *Although our main target of this experiment is to examine the reduction of generalization gap after noise injection, one may also find some other observations from the experiment results. For example, with noise injection, TRADES gets a better adversarial testing accuracy, while AT does not have significant change in this, which implies some potential differences between AT and TRADES to help people better understand why AT does not perform well.*

### C.2.2 Improving the attack error

From Corollary 1, an accurate attack leads to better algorithmic stability. To explore this in neural networks, since PGD-5 is more accurate than FGM, we use $\mathcal{L}_2$ adversarial training under different choices of $\epsilon$, and compare the adversarial training accuracy against the generalization gap for these two choices of attack method (i.e., FGM vs. PGD-5). To ensure the comparison is fair, we use FGM in testing data if the training uses FGM and use PGD-5 in testing data if the training uses PGD-5. The value of $\epsilon$ ranges from 0.25 to 4.0 to achieve different levels of adversarial training accuracy.

As shown in Figure 3, when the same level of adversarial training accuracy (70%~80%) is achieved, the generalization gap of FGM, in a worse-case scenario, can be much larger than PGD.

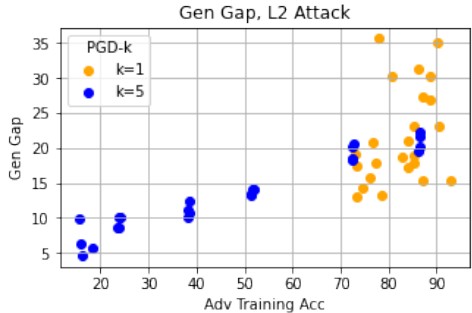

Figure 3: Comparison between FGM (i.e., PGD-1) and PGD-5 in adversarial training accuracy against generalization gap. The generalization gap using FGM is poor. Different values are obtained under different attack strength.

## D  Proofs

### D.1  Adversarial Training without Noise Injection

*Proof of Theorem 1.* The proof mainly follows Bassily et al. (2020) and we will transfer their worst-case scenario into the format of adversarial training.

Let $D = \min\{t, 1/\eta^2\} \leq d$, and $\nu > 0$, $K \geq \sqrt{D}$. Take linear loss functions and $y \equiv 0$. Denote $h$ as a smooth activation function to approximate ReLU, with $h = 0$ when $z < -\zeta$, and $h(z) = z$ when $z > \zeta$. Define $f$ as

$$f_\theta(x, z) = \begin{cases} \|h(\theta - x)\|_H & z < 0.5 - \lambda \\ r^\top \theta / K & z \geq 0.5 + \lambda \\ \frac{z - 0.5 + \lambda}{2\lambda} \|h(\theta - x)\|_H + \frac{0.5 + \lambda - z}{2\lambda} r^\top \theta / K & \text{otherwise} \end{cases}$$

where $\|h(\theta - x)\|_H$ is a smooth approximation of $\max\{0, \theta - x\}$, and $r$ is a vector with first $D$ elements as -1 and others as 0. The tuple $(x, z)$ represents the independent variables in the data.

Consider attack strength $\epsilon < 0.5$, then taking $z \in \{0, 1\}$, this attack strength is not strong enough to change whether $z$ is greater than 0.5 or not, and the attack will only change $x$.

In the first dataset $S_1$, $(x_1, z_1) = (\nu, ..., \nu, 1)$, and the others are $(x_i, z_i) = (\nu, ..., \nu, 0)$. In the second dataset $S_2$, all the samples are $(x_i, z_i) = (\nu, ..., \nu, 0)$.

Take initialization $\theta_1^{(0)} = \theta_2^{(0)} = \mathbf{0}$. If attack strength is small enough such that $\epsilon < \nu$, then $\theta_2^{(t)}$ is always $\mathbf{0}$.

To analyze $\theta_1^{(t)}$, for SGD, denote $i_t$ as the sample index at $t$th iteration. The value of $\theta_1^{(t)}$ keeps $\mathbf{0}$ before the first $(x_1, z_1)$ appears. For the first $t$ (denote as $t_0$) such that $i_t = 1$, since the function $f_\theta$ is not related to $x$ when $z = 1$, we have $\theta_1^{(t_0)} = -\eta r/K$. Taking $\nu < \eta/K$, for the next step $t_0 + 1$, if $i_{t_0+1}$ is not 1, then the attack will randomly select an element of $x$ to result in

$$\max_{x' \in B_2(x_{i_{t_0+1}}, \epsilon)} f_{\theta_1^{(t_0)}}(x', z_{i_{t_0+1}}) = \eta/K - \nu + \epsilon + o,$$

where we chose $h$ and $H$ properly so that its gradient has only minor difference with $\max\{0, \theta - x\}$, which is represented as $o$. The update of $\theta_1$ will be in the corresponding dimension chosen by attack, which has the same outcome as the nonsmooth function considered in Bassily et al. (2020) for clean training. Then the remaining proof follows the one in Bassily et al. (2020) directly.

The proof of GD is similar as SGD, as we take sufficiently large $K$ such that $\nu < \eta\|r\|/K$.

$\square$

*Proof of Theorem 5.* In the proof, we slightly change the assumption on $\kappa$ to

$$\|\nabla_\theta l(f_{\theta_1}(x_1), y) - \nabla_\theta l(f_{\theta_2}(x_2), y)\|^2 + \|\nabla_x l(f_{\theta_1}(x_1), y) - \nabla_x l(f_{\theta_2}(x_2), y)\|^2 \le \kappa^2(\|\theta_1 - \theta_2\|^2 + \|x_1 - x_2\|^2).$$

Define $z_i^1$ and $z_i^2$ as the correct attack of sample $(x_i, y_i)$ given the models $\theta_1^{(t)}$ and $\theta_2^{(t)}$. For SGD, we have

$$
\begin{aligned}
\Delta_{t+1} &\le \left\| \theta_1^{(t)} - \theta_2^{(t)} - \eta_t \left( \nabla_\theta l(f_{\theta_1^{(t)}}(\widehat{z}_{i_t}^1), y_{i_t}^1) - \nabla_\theta l(f_{\theta_2^{(t)}}(\widehat{z}_{i_t}^2), y_{i_t}^2) \right) \right\| \\
&\le \Delta_t + \eta_t \left\| \nabla_\theta l(f_{\theta_1^{(t)}}(\widehat{z}_{i_t}^1), y_{i_t}^1) - \nabla_\theta l(f_{\theta_2^{(t)}}(\widehat{z}_{i_t}^2), y_{i_t}^2) \right\| \\
&= \Delta_t + \eta_t \left\| \nabla_\theta l(f_{\theta_1^{(t)}}(\widehat{z}_{i_t}^1 - z_{i_t}^1 + z_{i_t}^1 - x_{i_t}^1 + x_{i_t}^1), y_{i_t}^1) - \nabla_\theta l(f_{\theta_2^{(t)}}(\widehat{z}_{i_t}^2 - z_{i_t}^2 + z_{i_t}^2 - x_{i_t}^2 + x_{i_t}^2), y_{i_t}^2) \right\| \\
&\le \Delta_t + 2\eta_t l \Delta\varepsilon + 2\eta_t L + \eta_t \left\| \nabla_\theta l(f_{\theta_1^{(t)}}(x_{i_t}^1), y_{i_t}^1) - \nabla_\theta l(f_{\theta_2^{(t)}}(x_{i_t}^2), y_{i_t}^2) \right\|.
\end{aligned}
$$

Therefore, given $\Delta_t$,

$$
\begin{aligned}
\mathbb{E}[\Delta_{t+1}|\Delta_t] &\le \Delta_t + 2\eta_t(\kappa\Delta\varepsilon + L) + \eta_t \mathbb{E}\left[ \left\| \nabla_\theta l(f_{\theta_1^{(t)}}(x_{i_t}^1), y_{i_t}^1) - \nabla_\theta l(f_{\theta_2^{(t)}}(x_{i_t}^2), y_{i_t}^2) \right\| \mathbf{1}\{(x_{i_t}^1, y_{i_t}^1) = (x_{i_t}^2, y_{i_t}^2)\} \right] \\
&\quad + \eta_t \mathbb{E}\left[ \left\| \nabla_\theta l(f_{\theta_1^{(t)}}(x_{i_t}^1), y_{i_t}^1) - \nabla_\theta l(f_{\theta_2^{(t)}}(x_{i_t}^2), y_{i_t}^2) \right\| \mathbf{1}\{(x_{i_t}^1, y_{i_t}^1) \ne (x_{i_t}^2, y_{i_t}^2)\} \right] \\
&\le \Delta_t + 2\eta_t(\kappa\Delta\varepsilon + L) + \eta_t \kappa \Delta_t \frac{n-1}{n} + \frac{\eta_t L}{n}.
\end{aligned}
$$

Since $\eta_t \le c/t$, denoting $t_0$ as the known first time that the $i_t$th sample in the two datasets are difference, we have

$$
\begin{aligned}
\mathbb{E}[\Delta_t|t_0] &\le \left(1 + \frac{c\kappa}{t}\right) \mathbb{E}[\Delta_t|t_0] + \frac{c}{t}\left(2\kappa\Delta\varepsilon + 2L + \frac{L}{n}\right) \\
&\le \sum_{\tau=t_0+1}^{t} \prod_{k=\tau+1}^{t} \left(1 + \frac{c\kappa}{k}\right) \frac{c}{\tau}\left(2\kappa\Delta\varepsilon + 2L + \frac{L}{n}\right) \\
&\le \sum_{\tau=t_0+1}^{t} \prod_{k=\tau+1}^{t} \exp\left(\frac{c\kappa}{k}\right) \frac{c}{\tau}\left(2\kappa\Delta\varepsilon + 2L + \frac{L}{n}\right) \\
&= \sum_{\tau=t_0+1}^{t} \exp\left(\sum_{k=\tau+1}^{t} \frac{c\kappa}{k}\right) \frac{c}{\tau}\left(2\kappa\Delta\varepsilon + 2L + \frac{L}{n}\right) \\
&\le \sum_{\tau=t_0+1}^{t} \exp\left(c\kappa \log(t/\tau)\right) \frac{c}{\tau}\left(2\kappa\Delta\varepsilon + 2L + \frac{L}{n}\right) \\
&= ct^{c\kappa}\left(2\kappa\Delta\varepsilon + 2L + \frac{L}{n}\right) \sum_{\tau=t_0+1}^{t} \tau^{-c\kappa-1} \\
&\le c\left(2\kappa\Delta\varepsilon + 2L + \frac{L}{n}\right)\left(\frac{t}{t_0}\right)^{c\kappa}.
\end{aligned}
$$

As a result, taking expectation w.r.t $t_0$, we have

$$\mathbb{E}[\Delta_t] \leq c\left(2\kappa\Delta\varepsilon + 2L + \frac{L}{n}\right)\mathbb{E}\left[\left(\frac{t}{t_0}\right)^{c\kappa}1\{t_0 \geq t_1\}\right] + 2rP(t_0 < t_1)$$

$$\leq c\left(2\kappa\Delta\varepsilon + 2L + \frac{L}{n}\right)\left(\frac{t}{t_1}\right)^{c\kappa}P(t_0 \geq t_1) + 2rP(t_0 < t_1).$$

Taking $t_1 = \Theta((nt^{c\kappa})^{\frac{1}{1+c\kappa}})$, it becomes

$$\mathbb{E}[\Delta_t] = O\left(\left(2\kappa\Delta\varepsilon + 2L + \frac{L}{n}\right)\left(\frac{t}{n}\right)^{\frac{c\kappa}{1+c\kappa}}\right).$$

For GD, since the different sample appears in the first iteration, we directly take $t_0 = 1$ in (6) and obtain the result. $\qquad\square$

## D.2  Adversarial Training with Noise Injection

We first present the formal statement of Lemma 1 as follows:

**Lemma 3.** *Assume Assumption 1 holds. Denote $L$ as the Lipschitz constant of $l(f_\theta(x), y_j)$ w.r.t. $\theta$ for any $x \in B_2(x_j, 2\epsilon)$ and all $1 \leq j \leq n$, and $\kappa$ as the Lipschitz constant of $\nabla_\theta l(f_\theta(x), y)$ w.r.t. $x$. Take $B$ as some function (specified later) of $(L, n, d, \kappa)$. Then, $(B, L, \kappa)$ is bounded by some finite $(B^*, L^*, \kappa^*)$ with probability tending 1, where the probability refers to the generation measure of $S = \{x_j, y_j\}_{j=1}^n$.*

*Assume the noise injected in data is zero-mean Gaussian with variance $(\xi_0^2/d)I_d$, and the noise injected in parameters is zero-mean Gaussian with variance $(\xi^2/d)I_d$ with $\xi = \xi_0 L^*$ and $\xi(d\log n) \to 0$. Denote $E(\theta + \delta, \widetilde{x}, y)$ as the event that $\nabla_\theta l(f_{\theta+\delta}(\widetilde{x} + A_\epsilon(f_{\theta+\delta}, \widetilde{x}, y)), y)$ is $B^*/\zeta$-Lipschitz. Then for the regression and classification tasks, there exists some $\zeta \ll \xi \to 0$ in $n$ such that, with probability tending to one over the generation of $S$, uniformly for all $\theta \in B_2(0, r)$,*

$$P(E^c(\theta + \delta, \widetilde{x}, y)|(x, y) \in S) = o(1).$$

Let $P(E^c|S) := \sup_{\theta \in B_2(0,r),(x,y)} P(E^c(\theta + \delta, \widetilde{x}, y)|(x, y) \in S)$ in what follows, for notation simplicity.

**Remark 8.** *The terms $r$, $L$, $\kappa$ are generic representations. For different loss functions and data dimension $d$, their values may change. In addition, the exact rate of $P(E^c|S)$ is affected by the value of $r, L, \kappa$ as well as $\xi_0, \zeta_0$. We postpone the details to the proof.*

In the following proofs regarding to Theorem 2 and 3, we use linear regression as an example. To be more specific, the three lemmas to be used in the main proof, Lemma 4, Lemma 5 and 6, provide some results w.r.t $E$ and $\mathbb{E}g$ for linear regression model. The proof for Theorem 2 and 3 directly utilize the results on $E$ and $\mathbb{E}g$ instead of any specific model. We provide the results of $E$ and $\mathbb{E}g$ for other models in the next section. Theorem 2 and 3 also hold after replacing Lemma 5 and 6 by these lemmas.

In terms of Lemma 3, it is a summary of results of $L$ and $P(E^c|S)$ over different models.

**Lemma 4.** *For linear regression, there exists some $(L^*, \kappa^*)$ such that, with probability tending to one over the choice of $S$, $L \leq L^*$ and $\kappa \leq \kappa^*$.*

*Proof of Lemma 4.* The gradient can be written as

$$\frac{1}{2}\nabla_\theta l(f_\theta(x), y) = x(x^\top \theta - y).$$

Then from the definition of the Lipschitz constant $L$, when taking $\delta_x$ such that $\delta_x \in B_2(0, 2\epsilon)$,

$$\frac{1}{2}L = \max_{\theta \in B_2(0,r), i \in [n], \delta_x} \|x_i + \delta_x\| \|(x_i + \delta_x)^\top \theta - y_i\| \leq (\max_i \|x_i\| + 2\epsilon)^2 r + (\max_i \|x_i\| |y_i| + 2\epsilon |y_i|).$$

In addition,

$$\frac{1}{2}\left\|\nabla_\theta l(f_\theta(x), y) - \nabla_\theta l(f_\theta(x + \delta_x), y)\right\|$$

$$= \frac{1}{2}\left\|x(x^\top\theta - y) - x((x + \delta_x)^\top\theta - y) - \delta_x((x + \delta_x)^\top\theta - y)\right\|$$

$$= \frac{1}{2}\left\|x\delta_x^\top\theta - \delta_x((x + \delta_x)^\top\theta - y)\right\|$$

$$= \frac{1}{2}\left\|x\delta_x^\top\theta - \delta_x(x^\top\theta - y) - \delta_x\delta_x^\top\theta\right\|$$

$$\leq \frac{1}{2}\left(\|x\|\|\theta\|\|\delta_x\| + \|\delta_x\|\|x^\top\theta - y\| + \|\delta_x\|^2\|\theta\|\right)$$

$$\leq \frac{1}{2}\left(\|x\|\|\theta\|\|\delta_x\| + \|\delta_x\|\|x^\top\theta - y\| + \|\delta_x\|2\epsilon\|\theta\|\right).$$

Thus for a given set of data $S$,

$$\frac{1}{2}\kappa = \max_{\theta \in B_2(0,r), i \in [n]}\left[(\|x_i\| + 2\epsilon)\|\theta\| + |x_i^\top\theta - y_i|\right] \leq 2(\max_i\|x_i\| + \epsilon)r + \max_i|y_i|.$$

From the distribution of $x$, we know that $\max_i\|x_i\| = O(\sqrt{d\log n})$ almost surely. In addition, $\mathbb{E}\|x\|\|y|$ and $\mathbb{E}|y|$ are finite, thus $\max_i\|x_i\|\|y_i|$ and $\max_i|y_i|$ are some functions of $n$ as well.

$\square$

**Lemma 5.** *For linear regression, denote $\zeta = L\zeta_0$ for some $\zeta_0/\xi_0 \to 0$. Denote $E(\theta, \delta, \widetilde{x}, y) = 1\{\|\theta + \delta\| \geq \zeta, |\widetilde{x}^\top(\theta + \delta) - y| \geq \zeta_0 r(d\log n)\}$, then $E = 1$ implies that $\nabla_\theta l(f_{\theta+\delta}(\widetilde{x}), y)$ is $B/\zeta_0$-Lipschitz. Uniformly for all $\theta$, with probability tending to one over the $n$ random samples, we have*

$$P(E^c(\theta, \delta, \widetilde{x}, y)|S) = o(1).$$

*Proof of Lemma 5.* We show that $E = 1$ implies that $\nabla_\theta l(f_{\theta+\delta}(\widetilde{x}), y)$ is $B/\zeta_0$-Lipschitz. The gradient of adversarial loss is

$$\frac{1}{2}g(\widetilde{x}, y, \theta) = \widetilde{x}(\widetilde{x}^\top(\theta + \delta) - y) + \epsilon^2(\theta + \delta) + \epsilon\frac{(\theta + \delta)}{\|\theta + \delta\|}|y - \widetilde{x}^\top(\theta + \delta)| - \epsilon\widetilde{x}\|(\theta + \delta)\|\mathrm{sgn}(y - \widetilde{x}^\top(\theta + \delta)).$$

When $\|\theta + \delta\| \geq \zeta$, we have for any $\theta'$,

$$\frac{1}{\|\theta + \delta - \theta'\|^2}\left\|\frac{\theta'}{\|\theta'\|} - \frac{\theta + \delta}{\|\theta + \delta\|}\right\|^2 = \frac{2}{\|\theta + \delta - \theta'\|^2} - \frac{2}{\|\theta + \delta - \theta'\|^2}\frac{(\theta + \delta)^\top\theta'}{\|\theta'\|\|\theta + \delta\|}.$$

Taking $\theta' \propto -(\theta + \delta)$, the above quantity is maximized. Therefore, taking $\theta' = -\alpha(\theta + \delta)$ for $\alpha > 0$,

$$\frac{1}{\|\theta + \delta - \theta'\|^2}\left\|\frac{\theta'}{\|\theta'\|} - \frac{\theta + \delta}{\|\theta + \delta\|}\right\|^2 \leq \frac{4}{\|\theta + \delta + \alpha(\theta + \delta)\|^2}$$

$$\leq \lim_{\alpha \to 0^+}\frac{4}{\|\theta + \delta + \alpha(\theta + \delta)\|^2}$$

$$\leq \frac{4}{\zeta^2}.$$

When $|y - \widetilde{x}^\top(\theta + \delta)| \geq \gamma$ for some $\gamma$, this implies that the nearest $\theta'$ such that $\operatorname{sgn}(y - \widetilde{x}^\top(\theta + \delta))$ gets changed satisfies $\|\theta' - (\theta + \delta)\| = \gamma/\|\widetilde{x}\|$. As a result,

$$\frac{1}{\|\theta + \delta - \theta'\|}\left\|\widetilde{x}\|\theta + \delta\|\operatorname{sgn}(y - \widetilde{x}^\top(\theta + \delta)) - \widetilde{x}\|\theta'\|\operatorname{sgn}(y - \widetilde{x}^\top\theta')\right\|$$

$$\leq \frac{1}{\|\theta + \delta - \theta'\|}\left\|\widetilde{x}\|\theta + \delta\|\operatorname{sgn}(y - \widetilde{x}^\top\theta') - \widetilde{x}\|\theta'\|\operatorname{sgn}(y - \widetilde{x}^\top\theta')\right\|$$

$$+ \frac{1}{\|\theta + \delta - \theta'\|}\left\|\widetilde{x}\|\theta + \delta\|\operatorname{sgn}(y - \widetilde{x}^\top(\theta + \delta)) - \widetilde{x}\|\theta + \delta\|\operatorname{sgn}(y - \widetilde{x}^\top\theta')\right\|$$

$$\leq \frac{\|\widetilde{x}\|\|\theta + \delta - \theta'\|}{\|\theta + \delta - \theta'\|} + \frac{\|\widetilde{x}\|\|\theta + \delta\|}{\|\theta + \delta - \theta'\|}\left|\operatorname{sgn}(y - \widetilde{x}^\top(\theta + \delta)) - \operatorname{sgn}(y - \widetilde{x}^\top\theta')\right|$$

$$\leq \|\widetilde{x}\| + \frac{2\|\widetilde{x}\|^2 r}{\gamma}.$$

Take $\gamma = \zeta_0 r \|\widetilde{x}\|^2$ in the above inequality to obtain $\sqrt{d \log n} + 2/\zeta_0$-Lipschitz.

Therefore the overall gradient is Lipschitz with

$$\kappa + 2\epsilon^2 + 8\epsilon/\zeta_0 + 2\epsilon\sqrt{d \log n} \tag{6}$$

which can be rewritten as $B/\zeta_0$ for some $B$.

Now we turn to bound the probability of $E^c$.

$$P(E^c(\theta, \delta, \widetilde{x}, y)|S) \leq P(\|\theta + \delta\| < \zeta) + P(|\widetilde{x}^\top(\theta + \delta) - y| < \zeta_0 r(d \log n)|S).$$

For any $\theta$, based on the distribution of $\delta$, we have

$$P(\|\theta + \delta\| < \zeta|\theta) = O\left(\left(\frac{\zeta}{\xi}\right)^d\right).$$

On the other hand,

$$P(|\widetilde{x}^\top(\theta + \delta) - y| < \zeta_0 r(d \log n)|S) = P(|(\widetilde{x} - x)^\top\theta + (\widetilde{x} - x)^\top\delta + (x^\top\theta - y) + x^\top\delta| < \zeta_0 r(d \log n)|S).$$

When $\|\theta\| > Cr$, from the distribution of $x^\top\delta$, $(\widetilde{x} - x)^\top\delta$, $(x^\top\theta - y)$, and $(\widetilde{x} - x)^\top\theta$, we have for any $(x, y, \theta)$,

$$P\left(|(\widetilde{x} - x)^\top\theta + (\widetilde{x} - x)^\top\delta + (x^\top\theta - y) + x^\top\delta| < \zeta_0 r(d \log n)\bigg| x, y, \theta\right)$$

$$= O\left(P\left(|x^\top\delta + (\widetilde{x} - x)^\top\theta| < \zeta_0 r(d \log n)\bigg| x\right)\right)$$

$$= O\left(\min\left(\frac{\zeta_0 r(d \log n)}{\|x\|\xi/\sqrt{d}}, \frac{\zeta_0 r(d \log n)}{\xi_0 r}, 1\right)\right).$$

From the distribution of $x$, with probability tending to one over the choice of $S$,

$$\mathbb{E}\left[\min\left(\frac{\zeta_0 r(d \log n)}{\|x\|\xi/\sqrt{d}}, \frac{\zeta_0 r(d \log n)}{\xi_0 r}, 1\right)\bigg| S\right]$$

$$\leq \mathbb{E}\left[\frac{\zeta_0 r(d \log n)}{\xi_0 r}1\{\|x\| \leq \zeta_0'\}\bigg| S\right] + \mathbb{E}\left[\frac{\zeta_0 r(d \log n)1\{\|x\| > \zeta_0'\}}{\|x\|\xi/\sqrt{d}}\bigg| S\right]$$

$$= O\left(\frac{\zeta_0 r(d \log n)}{\xi_0 r}(\zeta_0')^d + \frac{\zeta_0 r(d \log n)}{\zeta_0'\xi/\sqrt{d}}\right),$$

and take $\zeta_0' = (\xi_0 r\sqrt{d}/\xi)^{1/(d+1)}$ to reach the minimized upper bound as $O(\zeta_0 r(d \log n)(\xi_0 r\sqrt{d}/\xi)^{d/(d+1)})$.

When $\|\theta\| \leq Cr$, we first assume that $P(|x^\top \theta - y| \leq zr|S) = O(z) \; \forall z$ is correct to finish the main proof, and finally provide the proof of itself. From Assumption 1, we have $P(|x^\top \theta - y| \leq zr) = O(z)$. Since $\max \|x_i\| = O(\sqrt{d \log n})$ almost surely,

$$P\left(|(\tilde{x} - x)^\top \theta + (\tilde{x} - x)^\top \delta + (x^\top \theta - y) + x^\top \delta| < \zeta_0 r(d \log n) \Big| S\right)$$

$$\leq \; P\left(|x^\top \theta - y| < \zeta_0 r(d \log n) + |(\tilde{x} - x)^\top \theta + (\tilde{x} - x)^\top \delta + x^\top \delta| \Big| S\right)$$

$$= \; O\left(\frac{\zeta_0 r(d \log n) + L\xi_0 + \sqrt{d \log n}\xi}{r}\right) = O\left(\zeta_0(d \log n) + \xi_0 \frac{L}{r} + \frac{\sqrt{d \log n}\xi_0 L}{r}\right).$$

To conclude, with probability tending to one over the generation of $S$, we have

$$P(E^c|S) = O\left(\left(\frac{\zeta_0}{\xi_0}\right)^d + \zeta_0(d \log n)\left(\frac{r\sqrt{d}}{L}\right)^{\frac{d}{d+1}} + \xi_0 \frac{L}{r} + \frac{\sqrt{d \log n}\xi_0 L}{r}\right).$$

The last thing is to verify $P(|x^\top \theta - y| \leq \zeta_0 r|S) = O(\zeta_0)$ for any $\|\theta\| \leq r$. Using Bernstein inequality, we know that for any fixed $\theta$,

$$P\left(\sum_{i=1}^n 1\{|x_i^\top \theta - y_i| \leq \zeta_0 r\} - nP(|x^\top \theta - y| \leq \zeta_0 r) \geq t\right) \leq e^{-\frac{t^2}{n+t}}. \tag{7}$$

We construct some intervals and design a series of points in $B_2(0, r)$. For the interval $[-r, r]$, we equally divide it into $n^m$ sub-intervals and repeat this procedure on all the $d$ dimensions. Through this construction, there are $\Theta(n^{md})$ points in $B_2(0, r)$. Denote these points as $\alpha_i$ for $i = 1, ..., K$. A consequence of this construction is that, for any $\theta \in B_2(0, r)$, the nearest $\alpha_j$ to $\theta$ has distance less than $D = 2\sqrt{d}r/n^m$.

Taking $\{\alpha_j\}_{j=1,...,K}$ into (7), we obtain

$$P\left(\sup_{j \in [K]} \sum_{i=1}^n 1\{|x_i^\top \alpha_j - y_i| \leq \zeta_0 r\} - nP(|x^\top \alpha_j - y| \leq \zeta_0 r) \geq t\right) \leq Ke^{-\frac{t^2}{n+t}}.$$

For any $\theta$, denote $\alpha_k$ as the one in $\{\alpha_j\}_{j=1,...,K}$ such that $\|\theta - \alpha_k\|$ is minimized, then for a sample $(x_i, y_i)$,

$$|1\{|x_i^\top \theta - y_i| \leq \zeta_0 r\} - 1\{|x_i^\top \alpha_k - y_i| \leq \zeta_0 r\}|$$

$$= \; 1\{|x_i^\top \theta - y_i| \leq \zeta_0 r, |x_i^\top \alpha_k - y_i| > \zeta_0 r\} + 1\{|x_i^\top \theta - y_i| > \zeta_0 r, |x_i^\top \alpha_k - y_i| \leq \zeta_0 r\}$$

$$\leq \; 1\{|x_i^\top \alpha_k - y_i| - \|x_i\|\|\theta - \alpha_k\| \leq \zeta_0 r, |x_i^\top \alpha_k - y_i| > \zeta_0 r\}$$

$$+ 1\{|x_i^\top \alpha_k - y_i| + \|x_i\|\|\theta - \alpha_k\| > \zeta_0 r, |x_i^\top \alpha_k - y_i| \leq \zeta_0 r\}$$

$$\leq \; 1\{|x_i^\top \alpha_k - y_i| - \|x_i\|D \leq \zeta_0 r, |x_i^\top \alpha_k - y_i| > \zeta_0 r\}$$

$$+ 1\{|x_i^\top \alpha_k - y_i| + \|x_i\|D > \zeta_0 r, |x_i^\top \alpha_k - y_i| \leq \zeta_0 r\}$$

$$\leq \; 1\{\zeta_0 r - \|x_i\|D \leq |x_i^\top \alpha_k - y_i| \leq \zeta_0 r + \|x_i\|D\}.$$

Since $\max_i \|x_i\| = O(\sqrt{d \log n})$ almost surely, we can further expand the above formula into

$$1\{\zeta_0 r - \|x_i\|D \leq |x_i^\top \alpha_k - y_i| \leq \zeta_0 r + \|x_i\|D\}$$

$$\leq \; 1\{\zeta_0 r - Dc\sqrt{d \log n} \leq |x_i^\top \alpha_k - y_i| \leq \zeta_0 r + Dc\sqrt{d \log n}, \|x_i\| \leq c\sqrt{d \log n}\} + 1\{\|x_i\| > c\sqrt{d \log n}\}$$

$$\leq \; 1\{\zeta_0 r - Dc\sqrt{d \log n} \leq |x_i^\top \alpha_k - y_i| \leq \zeta_0 r + Dc\sqrt{d \log n}\} + 1\{\|x_i\| > c\sqrt{d \log n}\}.$$

As a result,

$$\left|\sum_{i=1}^n 1\{|x_i^\top \theta - y_i| \leq \zeta_0 r\} - \sum_{i=1}^n 1\{|x_i^\top \alpha_k - y_i| \leq \zeta_0 r\}\right|$$

$$\leq \; \sum_{i=1}^n 1\{\zeta_0 r - Dc\sqrt{d \log n} \leq |x_i^\top \alpha_k - y_i| \leq \zeta_0 r + Dc\sqrt{d \log n}\} + n1\{\max_i \|x_i\| > c\sqrt{d \log n}\},$$

where $n1\{\max_i \|x_i\| > c\sqrt{d\log n}\} = 0$ almost surely, and

$$P\left(\sup_j \sum_{i=1}^n 1\{|x_i^\top \alpha_j - y_i| \in \zeta_0 r \pm Dc\sqrt{d\log n}\} - nP\{|x^\top \alpha_j - y| \in \zeta_0 r \pm Dc\sqrt{d\log n}\} \geq t\right) \leq Ke^{-\frac{t^2}{n+t}}.$$

Consequently, rewrite $k$ as $k(\theta)$, we have

$$P\left(\sup_\theta \sum_{i=1}^n 1\{|x_i^\top \theta - y_i| \leq \zeta_0 r\} - nP(|x^\top \theta - y| \leq \zeta_0 r) \geq t\right)$$

$$\leq P\left(\sup_\theta \left[\sum_{i=1}^n 1\{|x_i^\top \alpha_{k(\theta)} - y_i| \leq \zeta_0 r\} - nP(|x^\top \alpha_{k(\theta)} - y| \leq \zeta_0 r)\right]\right.$$

$$+ \left[\sum_{i=1}^n 1\{|x_i^\top \alpha_{k(\theta)} - y_i| \in \zeta_0 r \pm Dc\sqrt{d\log n}\} - nP\{|x^\top \alpha_j - y| \in \zeta_0 r \pm Dc\sqrt{d\log n}\}\right]$$

$$+ nP\{|x^\top \alpha_{k(\theta)} - y| \in \zeta_0 r \pm Dc\sqrt{d\log n}\}$$

$$\left. + nP(|x^\top \alpha_{k(\theta)} - y| \leq \zeta_0 r) - nP(|x^\top \theta - y| \leq \zeta_0 r) \geq t\right)$$

$$\leq P\left(\sup_j \left[\sum_{i=1}^n 1\{|x_i^\top \alpha_j - y_i| \leq \zeta_0 r\} - nP(|x^\top \alpha_j - y| \leq \zeta_0 r)\right]\right.$$

$$+ \left[\sum_{i=1}^n 1\{|x_i^\top \alpha_j - y_i| \in \zeta_0 r \pm Dc\sqrt{d\log n}\} - nP\{|x^\top \alpha_j - y| \in \zeta_0 r \pm Dc\sqrt{d\log n}\}\right]$$

$$\left. + nP\{|x^\top \alpha_j - y| \in \zeta_0 r \pm Dc\sqrt{d\log n}\} + nP(|x^\top \alpha_j - y| \leq \zeta_0 r) \geq t\right)$$

$$\leq P\left(\sup_j \left[\sum_{i=1}^n 1\{|x_i^\top \alpha_j - y_i| \leq \zeta_0 r\} - nP(|x^\top \alpha_j - y| \leq \zeta_0 r)\right]\right.$$

$$\left. + nP\{|x^\top \alpha_j - y| \in \zeta_0 r \pm Dc\sqrt{d\log n}\} + nP(|x^\top \alpha_j - y| \leq \zeta_0 r) \geq \frac{t}{2}\right)$$

$$+ P\left(\sup_j \left[\sum_{i=1}^n 1\{|x_i^\top \alpha_j - y_i| \in \zeta_0 r \pm Dc\sqrt{d\log n}\} - nP\{|x^\top \alpha_j - y| \in \zeta_0 r \pm Dc\sqrt{d\log n}\}\right]\right.$$

$$\left. + nP\{|x^\top \alpha_j - y| \in \zeta_0 r \pm Dc\sqrt{d\log n}\} + nP(|x^\top \alpha_j - y| \leq \zeta_0 r) \geq \frac{t}{2}\right).$$

Denote $\gamma = \sup_j P\{|x^\top \alpha_j - y| \in \zeta_0 r \pm Dc\sqrt{d\log n}\} + P(|x^\top \alpha_j - y| \leq \zeta_0 r)$, then

$$P\left(\sup_{\theta \in B_2(0,r)} \frac{1}{n}\sum_{i=1}^n 1\{|x_i^\top \theta - y_i| \leq \zeta_0 r\} - P(|x^\top \theta - y| \leq \zeta_0 r) \geq t\right) \leq 2K\exp\left\{-\frac{n[(t/2 - \gamma)^+]^2}{1 + (t/2 - \gamma)^+}\right\}.$$

Recall that $K = \Theta(n^{md})$ and $D = 2\sqrt{d}r/n^m$, thus $\gamma = O\left(\sqrt{d}/n^m + \zeta_0\right)$. Taking $m$ as a constant such that $\zeta_0 \gg \sqrt{d}/n^m$, and $n\zeta_0$ grows polynomially in $n$, we have with probability tending to one over the generation of $S$, for any $\theta \in B_2(0,r)$

$$\frac{1}{n}\sum_{i=1}^n 1\{|x_i^\top \theta - y_i| \leq \zeta_0 r\} = O(\zeta_0).$$

$\square$

**Lemma 6.** *Under the same conditions as Lemma 5, with probability tending to one over the choice of $S$,*

$$\mathbb{E}[g(\widetilde{x}, y, \theta + \delta)^\top(\theta - \bar\theta)|S] \geq R_S(\theta) - R_S(\bar\theta) + O(\xi L^*).$$

*Proof of Lemma 6.* Since the adversarial loss is a convex function in both $\theta$ and $x$, and is smooth in $x$, we have

$$\mathbb{E}[g(\widetilde{x}, y, \theta + \delta)^\top (\theta - \bar{\theta})|S] \geq \mathbb{E}[l(f_{\theta+\delta}(\widetilde{x} + A_\epsilon(f, \widetilde{x}, y)), y)|S] - \mathbb{E}[l(f_{\bar{\theta}}(\widetilde{x} + A_\epsilon(f, \widetilde{x}, y)), y)|S].$$

To quantify the error introduced by $\widetilde{x}$, we have

$$\mathbb{E}[l(f_{\theta+\delta}(\widetilde{x} + A_\epsilon(f, \widetilde{x}, y)), y)|\delta, S]$$

$$= \mathbb{E}\left[(y - \widetilde{x}^\top(\theta + \delta))^2 + \epsilon^2\|\theta + \delta\|^2 + 2\epsilon\|\theta + \delta\|\,|y - \widetilde{x}^\top(\theta + \delta)|\,\Big|\,\delta, S\right]$$

$$\geq \mathbb{E}\left[(y - x^\top(\theta + \delta))^2 + ((\widetilde{x} - x)(\theta + \delta))^2 + \epsilon^2\|\theta + \delta\|^2 + 2\epsilon\|\theta + \delta\|\,|y - x^\top(\theta + \delta)| - 2\epsilon\|\theta + \delta\|\,|(\widetilde{x} - x)^\top(\theta + \delta)|\,\right.$$

$$= \mathbb{E}[l(f_{\theta+\delta}(x + A_\epsilon(f, x, y)), y)|\delta, S] + O(\xi_0^2 r^2) + O(\xi_0 r^2).$$

Similarly we can obtain a bound for $\mathbb{E}[l(f_{\bar{\theta}}(\widetilde{x} + A_\epsilon(f, \widetilde{x}, y)), y)|S]$.

Finally, from the distribution of $\delta$, we have

$$\mathbb{E}[l(f_{\theta+\delta}(x + A_\epsilon(f, x, y)), y)|S] = \mathbb{E}[l(f_\theta(x + A_\epsilon(f, x, y)), y)|S] + O(\xi L).$$

From the definition of $L^*$, we know that $r = O(L^*)$ and $L = O(L^*)$.

Consequently, aggregating all the above results, we have

$$\mathbb{E}[g(\widetilde{x}, y, \theta + \delta)^\top (\theta - \bar{\theta})|S] \geq \mathbb{E}[l(f_{\theta+\delta}(\widetilde{x} + A_\epsilon(f, \widetilde{x}, y)), y)|S] - \mathbb{E}[l(f_{\bar{\theta}}(\widetilde{x} + A_\epsilon(f, \widetilde{x}, y)), y)|S]$$

$$= \mathbb{E}[l(f_{\theta+\delta}(x + A_\epsilon(f, x, y)), y)|\delta, S] - \mathbb{E}[l(f_{\bar{\theta}}(x + A_\epsilon(f, x, y)), y)|\delta, S] + O(\xi L^*)$$

$$= \mathbb{E}[l(f_\theta(x + A_\epsilon(f, x, y)), y)|S] - \mathbb{E}[l(f_{\bar{\theta}}(x + A_\epsilon(f, x, y)), y)|\delta, S] + O(\xi L^*)$$

$$= R_S(\theta) - R_S(\bar{\theta}) + O(\xi L^*).$$

$\square$

*Proof of Lemma 2 and Theorem 2.* We use $(L, B, l)$ rather than $(L^*, B^*, l^*)$. The latter can is just an simple upper bound after obtain results regarding to the former one.

To show the stability of SGD, denoting $\Delta_t = \theta_1^{(t)} - \theta_2^{(t)}$, we have

$$\|\Delta_t\|^2 \leq \left\|\theta_1^{(t-1)} - \theta_2^{(t-1)} - \eta_t\left(\nabla_\theta l(f_{\theta_1^{(t-1)}+\delta}(\widehat{z}_{i_t}^1), y_{i_t}^1) - \nabla_\theta l(f_{\theta_2^{(t-1)}+\delta}(\widehat{z}_{i_t}^2), y_{i_t}^2)\right)\right\|^2$$

$$= \|\Delta_{t-1}\|^2 + \eta_t^2\left\|\nabla_\theta l(f_{\theta_1^{(t-1)}+\delta}(\widehat{z}_{i_t}^1), y_{i_t}^1) - \nabla_\theta l(f_{\theta_2^{(t-1)}+\delta}(\widehat{z}_{i_t}^2), y_{i_t}^2)\right\|^2$$

$$\qquad - 2\eta_t\Delta_{t-1}^\top\left(\nabla_\theta l(f_{\theta_1^{(t-1)}+\delta}(\widehat{z}_{i_t}^1), y_{i_t}^1) - \nabla_\theta l(f_{\theta_2^{(t-1)}+\delta}(\widehat{z}_{i_t}^2), y_{i_t}^2)\right).$$

Further,

$$\eta_t^2\left\|\nabla_\theta l(f_{\theta_1^{(t-1)}+\delta}(\widehat{z}_{i_t}^1 - z_{i_t}^1 + z_{i_t}^1, y_{i_t}^1) - \nabla_\theta l(f_{\theta_2^{(t-1)}+\delta}(\widehat{z}_{i_t}^2 - z_{i_t}^2 + z_{i_t}^2, y_{i_t}^2)\right\|^2$$

$$\leq \eta_t^2\left(2\kappa\Delta\varepsilon + \left\|\nabla_\theta l(f_{\theta_1^{(t-1)}+\delta}(z_{i_t}^1), y_{i_t}^1) - \nabla_\theta l(f_{\theta_2^{(t-1)}+\delta}(z_{i_t}^2), y_{i_t}^2)\right\|\right)^2,$$

$$\leq 2\eta_t^2\left(4\kappa^2\Delta\varepsilon^2 + \left\|\nabla_\theta l(f_{\theta_1^{(t-1)}+\delta}(z_{i_t}^1), y_{i_t}^1) - \nabla_\theta l(f_{\theta_2^{(t-1)}+\delta}(z_{i_t}^2), y_{i_t}^2)\right\|^2\right)$$

and

$$-2\eta_t\Delta_{t-1}^\top\left(\nabla_\theta l(f_{\theta_1^{(t-1)}+\delta}(\widehat{z}_{i_t}^1), y_{i_t}^1) - \nabla_\theta l(f_{\theta_2^{(t-1)}+\delta}(\widehat{z}_{i_t}^2), y_{i_t}^2)\right)$$

$$\leq 4\eta_t\|\Delta_{t-1}\|\kappa\Delta\varepsilon - 2\eta_t\Delta_{t-1}^\top\left(\nabla_\theta l(f_{\theta_1^{(t-1)}+\delta}(z_{i_t}^1), y_{i_t}^1) - \nabla_\theta l(f_{\theta_2^{(t-1)}+\delta}(z_{i_t}^2), y_{i_t}^2)\right).$$

Therefore, taking conditional expectation, we obtain

$$\mathbb{E}\left[\eta_t^2\left\|\nabla_\theta l(f_{\theta_1^{(t-1)}+\delta}(\widehat{z}_{i_t}^1 - z_{i_t}^1 + z_{i_t}^1, y_{i_t}^1) - \nabla_\theta l(f_{\theta_2^{(t-1)}+\delta}(\widehat{z}_{i_t}^2 - z_{i_t}^2 + z_{i_t}^2, y_{i_t}^2)\right\|^2\Big|\Delta_{t-1}\right]$$

$$-\mathbb{E}\left[2\eta_t\Delta_{t-1}^\top\left(\nabla_\theta l(f_{\theta_1^{(t-1)}+\delta}(\widehat{z}_{i_t}^1), y_{i_t}^1) - \nabla_\theta l(f_{\theta_2^{(t-1)}+\delta}(\widehat{z}_{i_t}^2), y_{i_t}^2)\right)\Big|\Delta_{t-1}\right]$$

$$\leq\ 8\eta_t^2\kappa^2\Delta\varepsilon^2 + 4\eta_t\|\Delta_{t-1}\|\kappa\Delta\varepsilon + 2\eta_t^2\mathbb{E}\left[\left\|\nabla_\theta l(f_{\theta_1^{(t-1)}+\delta}(z_{i_t}^1), y_{i_t}^1) - \nabla_\theta l(f_{\theta_2^{(t-1)}+\delta}(z_{i_t}^2), y_{i_t}^2)\right\|^2\Big|\Delta_{t-1}\right]$$

$$-2\eta_t\Delta_{t-1}^\top\mathbb{E}\left[\nabla_\theta l(f_{\theta_1^{(t-1)}+\delta}(z_{i_t}^1), y_{i_t}^1) - \nabla_\theta l(f_{\theta_2^{(t-1)}+\delta}(z_{i_t}^2), y_{i_t}^2)\Big|\Delta_{t-1}\right]$$

$$\leq\ 8\eta_t^2\kappa^2\Delta\varepsilon^2 + 4\eta_t\|\Delta_{t-1}\|\kappa\Delta\varepsilon + 2\eta_t^2(2L)^2\frac{1}{n}$$

$$+2\eta_t^2\mathbb{E}\left[\left\|\nabla_\theta l(f_{\theta_1^{(t-1)}+\delta}(z_{i_t}^1), y_{i_t}^1) - \nabla_\theta l(f_{\theta_2^{(t-1)}+\delta}(z_{i_t}^2), y_{i_t}^2)\right\|^2\Big|\Delta_{t-1}, (x_{i_t}^1, y_{i_t}^1) = (x_{i_t}^2, y_{i_t}^2)\right]$$

$$+2\eta_t\|\Delta_{t-1}\|(2L)\frac{1}{n}$$

$$-2\eta_t\Delta_{t-1}^\top\mathbb{E}\left[\nabla_\theta l(f_{\theta_1^{(t-1)}+\delta}(z_{i_t}^1), y_{i_t}^1) - \nabla_\theta l(f_{\theta_2^{(t-1)}+\delta}(z_{i_t}^2), y_{i_t}^2)\Big|\Delta_{t-1}, (x_{i_t}^1, y_{i_t}^1) = (x_{i_t}^2, y_{i_t}^2)\right]$$

$$\leq\ 8\eta_t^2\kappa^2\Delta\varepsilon^2 + 4\eta_t\|\Delta_{t-1}\|\kappa\Delta\varepsilon + 2\eta_t^2(2L)^2\frac{1}{n} + 2\eta_t^2(2L)^2 P(E^c|S)$$

$$+2\eta_t^2\mathbb{E}\left[\left\|\nabla_\theta l(f_{\theta_1^{(t-1)}+\delta}(z_{i_t}^1), y_{i_t}^1) - \nabla_\theta l(f_{\theta_2^{(t-1)}+\delta}(z_{i_t}^2), y_{i_t}^2)\right\|^2\Big|\Delta_{t-1}, (x_{i_t}^1, y_{i_t}^1) = (x_{i_t}^2, y_{i_t}^2), E\right]$$

$$+2\eta_t\|\Delta_{t-1}\|(2L)\frac{1}{n} + 2\eta_t\|\Delta_{t-1}\|(2L)P(E^c|S)$$

$$-2\eta_t\Delta_{t-1}^\top\mathbb{E}\left[\nabla_\theta l(f_{\theta_1^{(t-1)}+\delta}(z_{i_t}^1), y_{i_t}^1) - \nabla_\theta l(f_{\theta_2^{(t-1)}+\delta}(z_{i_t}^2), y_{i_t}^2)\Big|\Delta_{t-1}, (x_{i_t}^1, y_{i_t}^1) = (x_{i_t}^2, y_{i_t}^2), E\right].$$

Under $E$, since $l(f_{\theta_1^{(t-1)}+\delta}(z_{i_t}^1), y_{i_t}^1)$ is convex, following (A.1) of Hardt et al. (2016), we have

$$-2\eta_t\Delta_{t-1}^\top\mathbb{E}\left[\nabla_\theta l(f_{\theta_1^{(t-1)}+\delta}(z_{i_t}^1), y_{i_t}^1) - \nabla_\theta l(f_{\theta_2^{(t-1)}+\delta}(z_{i_t}^2), y_{i_t}^2)\Big|\Delta_{t-1}, (x_{i_t}^1, y_{i_t}^1) = (x_{i_t}^2, y_{i_t}^2), E\right]$$

$$\leq\ -2\eta_t\frac{\zeta}{B}\mathbb{E}\left[\left\|\nabla_\theta l(f_{\theta_1^{(t-1)}+\delta}(z_{i_t}^1), y_{i_t}^1) - \nabla_\theta l(f_{\theta_2^{(t-1)}+\delta}(z_{i_t}^2), y_{i_t}^2)\right\|^2\Big|\Delta_{t-1}, (x_{i_t}^1, y_{i_t}^1) = (x_{i_t}^2, y_{i_t}^2), E\right],$$

thus we obtain Lemma 2 for general choices of $\eta_t$.

$$\mathbb{E}[\|\theta_1^{(t)} - \theta_1^{(t)}\|^2|S]$$

$$\leq\ \left(1 + 2\eta_t^2\frac{B^2}{\zeta^2}1\{\eta_t \geq \frac{\zeta}{B}\}\right)\|\theta_1^{(t-1)} - \theta_2^{(t-1)}\|^2 + 8\eta_t^2\kappa^2\Delta\varepsilon^2 + 2\eta_t^2(2L)^2\frac{1}{n} + 2\eta_t^2(2L)^2 P(E^c|S)\qquad(8)$$

$$+4\eta_t\|\Delta_{t-1}\|\kappa\Delta\varepsilon + 2\eta_t\|\Delta_{t-1}\|(2L)\frac{1}{n} + 2\eta_t\|\Delta_{t-1}\|(2L)P(E^c|S).$$

When taking $\eta_t \leq \zeta/B$, we have

$$2\eta_t^2\mathbb{E}\left[\left\|\nabla_\theta l(f_{\theta_1^{(t-1)}+\delta}(z_{i_t}^1), y_{i_t}^1) - \nabla_\theta l(f_{\theta_2^{(t-1)}+\delta}(z_{i_t}^2), y_{i_t}^2)\right\|^2\Big|\Delta_{t-1}, (x_{i_t}^1, y_{i_t}^1) = (x_{i_t}^2, y_{i_t}^2), E\right]$$

$$-2\eta_t\Delta_{t-1}^\top\mathbb{E}\left[\nabla_\theta l(f_{\theta_1^{(t-1)}+\delta}(z_{i_t}^1), y_{i_t}^1) - \nabla_\theta l(f_{\theta_2^{(t-1)}+\delta}(z_{i_t}^2), y_{i_t}^2)\Big|\Delta_{t-1}, (x_{i_t}^1, y_{i_t}^1) = (x_{i_t}^2, y_{i_t}^2), E\right]$$

$$\leq\ 0.$$

Therefore,

$$\mathbb{E}\left[\eta_t^2 \left\|\nabla_\theta l(f_{\theta_1^{(t-1)}+\delta}(\hat{z}_{i_t}^1 - z_{i_t}^1 + z_{i_t}^1, y_{i_t}^1)) - \nabla_\theta l(f_{\theta_2^{(t-1)}+\delta}(\hat{z}_{i_t}^2 - z_{i_t}^2 + z_{i_t}^2, y_{i_t}^2))\right\|^2 \middle| \Delta_{t-1}\right]$$

$$-\mathbb{E}\left[2\eta_t \Delta_{t-1}^\top \left(\nabla_\theta l(f_{\theta_1^{(t-1)}+\delta}(\hat{z}_{i_t}^1), y_{i_t}^1) - \nabla_\theta l(f_{\theta_2^{(t-1)}+\delta}(\hat{z}_{i_t}^2), y_{i_t}^2)\right) \middle| \Delta_{t-1}\right]$$

$$\leq \quad 8\eta_t^2 \kappa^2 \Delta\varepsilon^2 + 2\eta_t^2(2L)^2 \frac{1}{n} + 2\eta_t^2(2L)^2 P(E^c)$$

$$+4\eta_t \|\Delta_{t-1}\| \kappa\Delta\varepsilon + 2\eta_t \|\Delta_{t-1}\|(2L)\frac{1}{n} + 2\eta_t \|\Delta_{t-1}\|(2L)P(E^c),$$

and

$$\mathbb{E}[\|\Delta_t\|^2 | \Delta_{t-1}] \quad \leq \quad \|\Delta_{t-1}\|^2 + 8\eta_t^2 \kappa^2 \Delta\varepsilon^2 + 2\eta_t^2(2L)^2 \frac{1}{n} + 2\eta_t^2(2L)^2 P(E^c)$$

$$+4\eta_t \|\Delta_{t-1}\| \kappa\Delta\varepsilon + 2\eta_t \|\Delta_{t-1}\|(2L)\frac{1}{n} + 2\eta_t \|\Delta_{t-1}\|(2L)P(E^c),$$

which leads to

$$\mathbb{E}^2\|\Delta_T\| \leq \mathbb{E}\|\Delta_T\| \sum_{t=t_0}^T \left[4\eta_t \kappa\Delta\varepsilon + \frac{4L\eta_t}{n} + 4\eta_t LP(E^c)\right] + \sum_{t=t_0}^T 8\eta_t^2 \kappa^2 \Delta\varepsilon^2 + 8\frac{L^2\eta_t^2}{n} + 8\eta_t^2 L^2 P(E^c).$$

Reordering some terms in the above inequality, we get

$$\left(\mathbb{E}\|\Delta_T\| - \sum_{t=t_0}^T \left[4\eta_t \kappa\Delta\varepsilon + \frac{4L\eta_t}{n} + 4\eta_t LP(E^c)\right]\right)^2 \leq \sum_{t=t_0}^T 8\eta_t^2 \kappa^2 \Delta\varepsilon^2 + 8\frac{L^2\eta_t^2}{n} + 8\eta_t^2 L^2 P(E^c),$$

so finally we obtain

$$\mathbb{E}\|\Delta_T\| \quad = \quad O\left(\left[\Delta\varepsilon + \sqrt{P(E^c)} + \sqrt{\frac{1}{n}}\right]\sqrt{\sum_{t=t_0}^T \eta_t^2}\right) + O\left(\left[\Delta\varepsilon + \frac{1}{n} + P(E^c)\right]\sum_{t=t_0}^T \eta_t\right)$$

$$= \quad O\left(\left[\sqrt{P(E^c)} + \sqrt{\frac{1}{n}}\right]\sqrt{\sum_{t=t_0}^T \eta_t^2}\right) + O\left(\left[\Delta\varepsilon + \frac{1}{n} + P(E^c)\right]\sum_{t=t_0}^T \eta_t\right).$$

The proof for GD is similar. $\qquad\qquad\square$

*Proof of Theorem 3.* We first assume the analytical solution of attack exists and there is no attack error in linear regression problem, then discussing how to consider the attack error and for the other loss functions.

The updating rule of SGD leads to

$$\|\theta^{(t)} - \bar{\theta}\|^2 \leq \|\theta^{(t-1)} - \bar{\theta} - \eta_t g_t\|^2 \leq \|\theta^{(t-1)} - \bar{\theta}\|^2 - 2\eta_t g_t^\top (\theta^{(t-1)} - \bar{\theta}) + \eta_t^2 L^2.$$

Taking expectation and move some terms, it becomes

$$\mathbb{E}g_t^\top(\theta^{(t-1)} - \bar{\theta}) \leq \frac{1}{2\eta_t}\mathbb{E}\|\theta^{(t-1)} - \bar{\theta}\|^2 - \frac{1}{2\eta_t}\mathbb{E}\|\theta^{(t)} - \bar{\theta}\|^2 + \frac{1}{2}\eta_t L^2.$$

Taking average over $t = 1$ to $T$, we have

$$\frac{1}{T}\mathbb{E}\left[\sum_{t=1}^T g_t^\top(\theta^{(t-1)} - \bar{\theta})\right] \quad \leq \quad \frac{1}{2T}\mathbb{E}\left[\sum_{t=1}^T \frac{1}{\eta_t}\|\theta^{(t-1)} - \bar{\theta}\|^2 - \frac{1}{\eta_t}\|\theta^{(t)} - \bar{\theta}\|^2\right] + \frac{L^2}{2T}\sum_{t=1}^T \eta_t$$

$$= \quad \frac{\mathbb{E}\|\theta^{(0)} - \bar{\theta}\|^2}{2\eta_1 T} - \frac{\mathbb{E}\|\theta^{(T)} - \bar{\theta}\|^2}{2\eta_T T}$$

$$+\frac{1}{2T}\mathbb{E}\left[\sum_{t=1}^{T-1}\left(\frac{1}{\theta_{t+1}} - \frac{1}{\theta_t}\right)\|\theta^{(t)} - \bar{\theta}\|^2\right] + \frac{L^2}{2T}\sum_{t=1}^T \eta_t.$$

Finally, since $R_S$ is a convex function, based on Lemma 6, we have

$$\frac{1}{T}\mathbb{E}\left[\sum_{t=1}^{T}g_t^\top(\theta^{(t-1)}-\bar{\theta})\right] \geq \frac{1}{T}\mathbb{E}\sum_{t=1}^{T}R_S(\theta^{(t)}) - R_S(\bar{\theta}) + O(\xi L^*) \geq \mathbb{E}\left[\min_{t=1,\dots,T}R_S(\theta^{(t)}) - R_S(\bar{\theta})\right] + O(\xi L^*).$$

The above proof assumes that attack has no error. To count for the attack error, since $\nabla R_S$ is Lipschitz in $x$, denoting $\widehat{g}_t$ as the gradient approximated, then $\widehat{g}_t = g_t + O(\kappa\Delta\varepsilon)$. So an additional $O(\kappa r\Delta\varepsilon)$ is introduced.

Since fixing the $n$ samples, taking expectation in SGD is reduced to GD, the above result also holds. $\qquad\square$

### D.3 Lemmas for Other Loss

**Lemma 7** (Smoothed Hinge Loss). *For smoothed hinge loss, $H(x)$ is defined as a strictly monotone function in $x$ with $H(x) = 1$ when $x \geq 1$ and $H(x) = 0$ when $x \leq -1$, and $xH(x)$ is convex. The derivative $H'$ satisfies $H'(-1) = H'(1) = 0$, and $H''$ is finite. define $E(\theta, \delta, \widetilde{x}, y) = 1\{\|\theta + \delta\| \geq \zeta\}$, then $E = 1$ implies that $g_t$ is Lipschitz with $B/\max(h, \zeta)$. Assume $h$ is a fixed constant. In addition, when $r/(\sqrt{d\log n}) \to 0$, with probability tending to one over the choice of $S$,*

$$\mathbb{E}[g(\widetilde{x}, y, \theta + \delta)^\top(\theta - \bar{\theta})|S] \geq R_S(\theta) - R_S(\bar{\theta}) + O(\xi L^*).$$

*Proof of Lemma 7.* Given $(\theta, x, y)$, we have

$$l(f_\theta(x), y) = (1 - y(x^\top\theta))H\left(\frac{1 - yx^\top\theta}{h}\right),$$

for $y \in \{\pm 1\}$. Thus the attack is

$$A = \begin{cases} -y\epsilon\frac{\theta}{\|\theta\|} & \text{if } 1 - y(x^\top\theta) > -\epsilon \\ \text{any } z \in B_2(x, \epsilon) & \text{otherwise} \end{cases}.$$

The adversarial risk becomes

$$l(f_\theta(x + A), y) = (1 - y((x + A)^\top\theta))H\left(\frac{1 - y(x + A)^\top\theta}{h}\right),$$

and the gradient becomes

$$g(x, y, \theta) = -y(x + A)\left[H\left(1 - y(x^\top\theta) + \epsilon\|\theta\|\right) + \frac{(1 - y(x^\top\theta) + \epsilon\|\theta\|)}{h}H'\left(\frac{1 - y(x^\top\theta) + \epsilon\|\theta\|}{h}\right)\right]$$

Since $H$ and $H'$ are differentiable, for any $g$, when $\|\theta\| \geq \zeta$, for any other $\theta' \neq \mathbf{0}$.

$$\|g(x, y, \theta) - g(x, y, \theta')\| \leq \frac{B}{\zeta}\|\theta - \theta'\|.$$

In terms of the expectation of $g(\widetilde{x}, y, \theta + \delta)$, since $l(f_{\theta+\delta}(\widetilde{x} + A_\epsilon(f, \widetilde{x}, y)), y)$ is convex, we have

$$\mathbb{E}g(\widetilde{x}, y, \theta + \delta)^\top(\theta - \bar{\theta}) \geq \mathbb{E}l(f_{\theta+\delta}(\widetilde{x} + A_\epsilon(f, \widetilde{x}, y)), y) - \mathbb{E}l(f_{\bar{\theta}}(\widetilde{x} + A_\epsilon(f, \widetilde{x}, y)), y).$$

Further, for any $(x, y, \theta)$,

$$\begin{aligned}
&\mathbb{E}l(f_{\theta+\delta}(\widetilde{x} + A_\epsilon(f, \widetilde{x}, y)), y) \\
=~& \mathbb{E}\left[(1 - y(\widetilde{x}^\top(\theta + \delta)) + \epsilon\|\theta + \delta\|)H\left(\frac{(1 - y(\widetilde{x}^\top(\theta + \delta)) + \epsilon\|\theta + \delta\|)}{h}\right)\right] \\
=~& (1 - y(x^\top\theta) + \epsilon\|\theta\|)H\left(\frac{(1 - y(x^\top(\theta + \delta)) + \epsilon\|\theta\|)}{h}\right) \\
&+ \mathbb{E}\left[(1 - y(\widetilde{x}^\top(\theta + \delta)) + \epsilon\|\theta + \delta\|)\left(H\left(\frac{(1 - y(\widetilde{x}^\top(\theta + \delta)) + \epsilon\|\theta + \delta\|)}{h}\right) - H\left(\frac{(1 - y(x^\top\theta) + \epsilon\|\theta\|)}{h}\right)\right)\right] \\
=~& l(f_\theta(x + A_\epsilon(f, x, y)), y) \\
&+ \mathbb{E}\left[(1 - y(\widetilde{x}^\top(\theta + \delta)) + \epsilon\|\theta + \delta\|)\left(H\left(\frac{(1 - y(\widetilde{x}^\top(\theta + \delta)) + \epsilon\|\theta + \delta\|)}{h}\right) - H\left(\frac{(1 - y(x^\top\theta) + \epsilon\|\theta\|)}{h}\right)\right)\right].
\end{aligned}$$

Furthermore,

$$
\begin{aligned}
\left| (1 - y(\widetilde{x}^\top(\theta + \delta)) + \epsilon\|\theta + \delta\|) \right| &\leq \left| (1 - y(x^\top(\theta)) + \epsilon\|\theta\|) \right| + \epsilon\|\delta\| + \left| y((\widetilde{x} - x)^\top(\theta + \delta)) \right| + \left| y(x^\top\delta) \right| \\
&\leq L^* + \epsilon\|\delta\| + \left| (\widetilde{x} - x)^\top(\theta + \delta) \right| + \left| x^\top\delta \right| \\
&= L^*(1 + o_p(1)).
\end{aligned}
$$

From the definition of $H$, we have

$$
\begin{aligned}
& H\left( \frac{(1 - y(\widetilde{x}^\top(\theta + \delta))) + \epsilon\|\theta + \delta\|}{h} \right) - H\left( \frac{(1 - y(x^\top\theta)) + \epsilon\|\theta\|}{h} \right) \\
=\ & \left[ \frac{-y\widetilde{x}^\top(\theta + \delta) + yx^\top\theta + \epsilon\|\theta + \delta\| - \epsilon\|\theta\|}{h} \right] H'\left( \frac{(1 - y(x^\top\theta)) + \epsilon\|\theta\|}{h} \right) \\
& + \Bigg\{ H\left( \frac{(1 - y(\widetilde{x}^\top(\theta + \delta))) + \epsilon\|\theta + \delta\|}{h} \right) - H\left( \frac{(1 - y(x^\top\theta)) + \epsilon\|\theta\|}{h} \right) \\
& \quad - \left[ \frac{-y\widetilde{x}^\top(\theta + \delta) + yx^\top\theta + \epsilon\|\theta + \delta\| - \epsilon\|\theta\|}{h} \right] H'\left( \frac{(1 - y(x^\top\theta)) + \epsilon\|\theta\|}{h} \right) \Bigg\} \\
=\ & O_p(\xi + r\xi_0).
\end{aligned}
$$

Consequently, when $r / \max_i \|x_i\| \to 0$, $r\xi_0 = O(\xi L^*)$, and

$$
\mathbb{E}l(f_{\theta + \delta}(\widetilde{x} + A_\epsilon(f, \widetilde{x}, y)), y) = l(f_\theta(x + A_\epsilon(f, x, y)), y) + O(\xi L^*). \tag{9}
$$

$\square$

**Lemma 8.** *For Logistic regression, denote $E(\theta, \delta, \widetilde{x}, y) = 1\{\|\theta + \delta\| \geq \zeta\}$. Then $E = 1$ implies that $\nabla_{\theta + \delta} l(f_{\theta + \delta}(\widetilde{x} + A), y)$ is $1/\zeta_0$-Lipschitz.*

*In addition,*

$$
\mathbb{E}g(\widetilde{x}, y, \theta + \delta)^\top(\theta - \bar{\theta}) \geq R_S(\theta) - R_S(\bar{\theta}) + O(\xi L^*).
$$

*Proof of Lemma 8.* For each data $(x, y)$, $l(f(x, \theta), y) = -1\{y = 1\}\log(p(x^\top\theta)) - 1\{y = -1\}\log(1 - p(x^\top\theta))$, where

$$
p(x^\top\theta) = \frac{1}{1 + e^{-x^\top\theta}}.
$$

Taking gradient w.r.t $\theta$, we obtain

$$
\nabla_\theta l(f_\theta(x), y) = -1\{y = 1\}\frac{xp'(x^\top\theta)}{p(x^\top\theta)} + 1\{y = -1\}\frac{xp'(x^\top\theta)}{1 - p(x^\top\theta)},
$$

where

$$
p'(x^\top\theta) = \frac{e^{-x^\top\theta}}{(1 + e^{-x^\top\theta})^2}.
$$

When $\theta \neq \mathbf{0}$, the attack is

$$
A = \begin{cases} -\epsilon\frac{\theta}{\|\theta\|} & y = 1 \\ \epsilon\frac{\theta}{\|\theta\|} & y = -1 \end{cases}, \tag{10}
$$

thus

$$
\nabla_\theta l(f_\theta(x + A), y) = -1\{y = 1\}\frac{(x - \frac{\epsilon\theta}{\|\theta\|})p'(x^\top\theta - \epsilon\|\theta\|)}{p(x^\top\theta - \epsilon\|\theta\|)} + 1\{y = -1\}\frac{(x + \frac{\epsilon\theta}{\|\theta\|})p'(x^\top\theta + \epsilon\|\theta\|)}{1 - p(x^\top\theta + \epsilon\|\theta\|)}.
$$

The above representation indicates that $\|\theta\| \geq \zeta$ implies $\nabla_\theta l(f_\theta(x + A), y)$ is $B/\zeta$-Lipschitz.

In terms of $\mathbb{E}g(\widetilde{x}, y, \theta + \delta)^\top(\theta - \bar{\theta})$, since $l(f_{\theta + \delta}(\widetilde{x} + A_\epsilon(f, \widetilde{x}, y)), y)$ is convex, we have

$$
\mathbb{E}g(\widetilde{x}, y, \theta + \delta)^\top(\theta - \bar{\theta}) \geq \mathbb{E}l(f_{\theta + \delta}(\widetilde{x} + A_\epsilon(f, \widetilde{x}, y)), y) - \mathbb{E}l(f_{\bar{\theta}}(\widetilde{x} + A_\epsilon(f, \widetilde{x}, y)), y).
$$

The remaining proof is similar as in Lemma 6 and Lemma 7. $\square$

# E  $\mathcal{L}_\infty$ Attack in Adversarial Training

As mentioned in the proof of Theorem 2 and 3, it only requires some conditions w.r.t Lipschitz continuous of the gradient as well as $\mathbb{E}g^\top(\theta - \widetilde{\theta})$. Therefore, we provide results of $\mathcal{L}_\infty$ adversarial training for linear regression setup.

In addition, based on the following lemmas, the value of $(L, L^*, \kappa, \kappa^*)$ is different from those in $\mathcal{L}_2$ adversarial training. Furthermore, the value of $B$ and $B^*$ are also enlarged for $\mathcal{L}_\infty$ adversarial training.

**Lemma 9.** *For linear regression, there exists some $(L^*, \kappa^*)$ such that, with probability tending to one over the choice of $S$, $L \leq L^*$ and $\kappa \leq \kappa^*$.*

*Proof of Lemma 9.*

$$\frac{1}{2}\nabla_\theta l(f_\theta(x), y) = x(x^\top\theta - y).$$

Then from the definition of $L$, when $\|\delta_x\|_\infty \leq 2\epsilon$, $\|\delta_x\| \leq 2\sqrt{d}\epsilon$.

$$\frac{1}{2}L = \max_{\theta \in B_2(0,r), i\in[n], \delta_x} \|x_i + \delta_x\|\|(x_i + \delta_x)^\top\theta - y_i\| \leq (\max_i \|x_i\| + 2\sqrt{d}\epsilon)^2 r + (\max_i \|x_i\|\|y_i\| + 2\sqrt{d}\epsilon|y_i|).$$

In addition,

$$\frac{1}{2}\|\nabla_\theta l(f_\theta(x), y) - \nabla_\theta l(f_\theta(x + \delta_x), y)\|$$

$$= \frac{1}{2}\|x(x^\top\theta - y) - x((x + \delta_x)^\top\theta - y) - \delta_x((x + \delta_x)^\top\theta - y)\|$$

$$= \frac{1}{2}\|x\delta_x^\top\theta - \delta_x((x + \delta_x)^\top\theta - y)\|$$

$$= \frac{1}{2}\|x\delta_x^\top\theta - \delta_x(x^\top\theta - y) - \delta_x\delta_x^\top\theta\|$$

$$\leq \frac{1}{2}\left(\|x\|\|\theta\|\|\delta_x\| + \|\delta_x\|\|x^\top\theta - y\| + \|\delta_x\|^2\|\theta\|\right)$$

$$\leq \frac{1}{2}\left(\|x\|\|\theta\|\|\delta_x\| + \|\delta_x\|\|x^\top\theta - y\| + \|\delta_x\|2\sqrt{d}\epsilon\|\theta\|\right).$$

Thus for a given set of data $S$,

$$\frac{1}{2}\kappa = \max_{\theta \in B_2(0,r), i\in[n]}\left[(\|x_i\| + 2\sqrt{d}\epsilon)\|\theta\| + |x_i^\top\theta - y_i|\right] \leq 2(\max_i \|x_i\| + \sqrt{d}\epsilon)r + \max_i |y_i|.$$

From the distribution of $x$, we know that $\max_i \|x_i\| = O(\sqrt{d\log n})$ almost surely. In addition, $\mathbb{E}\|x\|\|y\|$ and $\mathbb{E}|y|$ are finite, thus $\max_i \|x_i\|\|y_i\|$ and $\max_i |y_i|$ are some functions of $n$ as well.  □

**Lemma 10.** *For linear regression, denote $\zeta = L\zeta_0$ for some $\zeta_0/\xi_0 \to 0$. Denote $E(\theta, \delta, \widetilde{x}, y) = 1\{\min_j |\theta+\delta|_j \geq \zeta/\sqrt{d}, |\widetilde{x}^\top(\theta+\delta) - y| \geq \zeta_0 r(d\log n)\}$, then $E = 1$ implies that $\nabla_\theta l(f_{\theta+\delta}(\widetilde{x}), y)$ is $\sqrt{d}/\zeta_0$-Lipschitz. Then uniformly for all $\theta$, with probability tending to one over the $n$ random samples, we have*

$$P(E^c(\theta, \delta, \widetilde{x}, y)|S) = o(1).$$

*Proof of Lemma 10.* We show that $E = 1$ implies that $\nabla_\theta l(f_{\theta+\delta}(\widetilde{x}), y)$ is $1/\zeta_0$-Lipschitz. The gradient of adversarial loss is

$$\frac{1}{2}g(\widetilde{x}, y, \theta) = \widetilde{x}(\widetilde{x}^\top(\theta+\delta) - y) + \epsilon^2\|\theta+\delta\|_1\text{sgn}(\theta+\delta) + \epsilon\text{sgn}(\theta+\delta)|y - \widetilde{x}^\top(\theta+\delta)|$$

$$-\epsilon\widetilde{x}\|(\theta+\delta)\|_1\text{sgn}(y - \widetilde{x}^\top(\theta+\delta)).$$

When $\min_j |\theta+\delta|_j \geq \zeta/\sqrt{d}$, we have for any $\theta'$,

$$\frac{1}{\|\theta+\delta-\theta'\|^2}\|\text{sgn}(\theta+\delta) - \text{sgn}(\theta')\|^2 \leq \lim_{\alpha\to 0^+}\frac{d}{\|\theta+\delta+\alpha(\theta+\delta)\|^2} \leq \frac{d}{\zeta^2}.$$

When $|y - \widetilde{x}^\top(\theta + \delta)| \geq \gamma$ for some $\gamma$, this implies that the nearest $\theta'$ such that $\text{sgn}(y - \widetilde{x}^\top(\theta + \delta))$ gets changed satisfies $\|\theta' - (\theta + \delta)\| = \gamma/\|\widetilde{x}\|$. As a result,

$$\frac{1}{\|\theta + \delta - \theta'\|} \left\| \widetilde{x}\|\theta + \delta\|\text{sgn}(y - \widetilde{x}^\top(\theta + \delta)) - \widetilde{x}\|\theta'\|\text{sgn}(y - \widetilde{x}^\top\theta') \right\|$$

$$\leq \quad \frac{1}{\|\theta + \delta - \theta'\|} \left\| \widetilde{x}\|\theta + \delta\|_1 \text{sgn}(y - \widetilde{x}^\top\theta') - \widetilde{x}\|\theta'\|_1 \text{sgn}(y - \widetilde{x}^\top\theta') \right\|$$

$$+ \frac{1}{\|\theta + \delta - \theta'\|} \left\| \widetilde{x}\|\theta + \delta\|_1 \text{sgn}(y - \widetilde{x}^\top(\theta + \delta)) - \widetilde{x}\|\theta + \delta\|_1 \text{sgn}(y - \widetilde{x}^\top\theta') \right\|$$

$$\leq \quad \frac{\|\widetilde{x}\|\sqrt{d}\|\theta + \delta - \theta'\|}{\|\theta + \delta - \theta'\|} + \frac{\|\widetilde{x}\|\|\theta + \delta\|_1}{\|\theta + \delta - \theta'\|} \left| \text{sgn}(y - \widetilde{x}^\top(\theta + \delta)) - \text{sgn}(y - \widetilde{x}^\top\theta') \right|$$

$$\leq \quad \|\widetilde{x}\| + \frac{2\|\widetilde{x}\|^2\sqrt{d}}{\gamma}.$$

Take $\gamma = \zeta_0 r\|\widetilde{x}\|^2$ in the above inequality to obtain $\|\widetilde{x}\| + 2\sqrt{d}/\zeta_0$-Lipschitz.

Now we turn to bound the probability of $E^c$.

$$P(E^c(\theta, \delta, \widetilde{x}, y)|S) \leq P(\min_j |\theta + \delta|_j < \zeta/\sqrt{d}) + P(|\widetilde{x}^\top(\theta + \delta) - y| < \zeta_0 r(d \log n)|S)$$

For any $\theta$, we have

$$P(\min_j |\theta + \delta|_j < \zeta/\sqrt{d}|\theta) = O\left(1 - \left(1 - \frac{\zeta}{\xi}\right)^d\right) = O\left(\frac{d\zeta}{\xi}\right).$$

The remaining steps follows the same as in Lemma 5.

$\square$

**Lemma 11.** *Under the same conditions as Lemma 5, with probability tending to one over the set of $n$ random samples,*

$$\mathbb{E}g(\widetilde{x}, y, \theta + \delta)^\top(\theta - \bar\theta) \geq R_S(\theta) - R_S(\bar\theta) + O(\xi L^*).$$

*Proof of Lemma 11.* Since the adversarial loss is a convex function in both $\theta$ and $x$, and is smooth in $x$, we have

$$\mathbb{E}g(\widetilde{x}, y, \theta + \delta)^\top(\theta - \bar\theta) \geq \mathbb{E}l(f_{\theta+\delta}(\widetilde{x} + A_\epsilon(f, \widetilde{x}, y)), y) - \mathbb{E}l(f_{\bar\theta}(\widetilde{x} + A_\epsilon(f, \widetilde{x}, y)), y).$$

To quantify the error introduced by $\widetilde{x}$, we have

$$\mathbb{E}[l(f_{\theta+\delta}(\widetilde{x} + A_\epsilon(f, \widetilde{x}, y)), y)|\delta]$$

$$= \quad \mathbb{E}\left[(y - \widetilde{x}^\top(\theta + \delta))^2 + \epsilon^2\|\theta + \delta\|_1^2 + 2\epsilon\|\theta + \delta\|_1|y - \widetilde{x}^\top(\theta + \delta)|\Big|\delta\right]$$

$$\geq \quad \mathbb{E}\left[(y - x^\top(\theta + \delta))^2 + ((\widetilde{x} - x)(\theta + \delta))^2 + \epsilon^2\|\theta + \delta\|_1^2 + 2\epsilon\|\theta + \delta\|_1|y - x^\top(\theta + \delta)| - 2\epsilon\|\theta + \delta\|_1(\widetilde{x} - x)^\top(\theta + \delta)\right.$$

$$= \quad \mathbb{E}[l(f_{\theta+\delta}(x + A_\epsilon(f, x, y)), y)|\delta] + O(\xi_0^2 r^2) + O(\xi_0 r^2\sqrt{d}).$$

Since $l$ is square loss and $f_\theta$ is the linear model, we have $r = O(L^*/\sqrt{d})$, thus

$$\mathbb{E}[l(f_{\theta+\delta}(\widetilde{x} + A_\epsilon(f, \widetilde{x}, y)), y)|\delta] = \mathbb{E}[l(f_{\theta+\delta}(x + A_\epsilon(f, x, y)), y)|\delta] + O(\xi_0(L^*)^2/\sqrt{d}).$$

Finally, from the definition of $L$, we have

$$\mathbb{E}l(f_{\theta+\delta}(x + A_\epsilon(f, x, y)), y) = \mathbb{E}l(f_\theta(x + A_\epsilon(f, x, y)), y) + O(\xi L^*).$$

$\square$