# OpenReview forum: "On the Algorithmic Stability of Adversarial Training"
_NeurIPS.cc/2021/Conference — NeurIPS 2021 Poster_

### Official Review · Reviewer_6caz · 2021-07-15

**Rating:** 5
**Confidence:** 4

**Summary:**

This paper studies the theory of adversarial examples from the angle of algorithmic stability. More precisely, the authors characterize the algorithmic stability of the now famous adversarial training scheme. Their main findings can be summarized as follows.
1) Even if the loss function utilized during adversarial training is smooth, the adversarial perturbation applied to the input image causes some non-differentiability issues which impede guaranteeing a good algorithmic stability in the worst-case scenario.
2) The numerical approximation error of adversarial attacks used during training can have an impact on the algorithmic stability of the scheme.
3) Injecting noise to the learning procedure helps to make the loss function smooth again; hence restoring the algorithmic stability with high probability.

**Limitations And Societal Impact:**

As this is mostly a theory paper, I do not see any potential negative social impact.

**Main Review:**

Relevance:
The problem of designing models that are robust against adversarial attacks is a burning issue both in academia and in the industry right now. As adversarial training is one of the most efficient strategies against this problem, it is important to better understand this technique from a theoretical standpoint. Hence, the main focus of this paper is completely relevant to the machine learning community.

Clarity:
I had some troubles following some parts of the paper. I feel like the overall point could be much more convincing if the author were to reorganize and focus their contributions, at least in the main paper. For example, Lemma 1 seems overcomplicated in its current form because the authors wanted to encapsulate several Lemmas at once (as explained in the appendix). To clarify the contributions and to make the paper more readable, I would suggest that the authors focus on presenting results for the linear regression and leave similar findings with other models to the appendix. Note that results are actually demonstrated in the appendix only for the regression case anyway, and extensions are left to the reader. Similarly,  Algorithm 1 (while it is quite straightforward) seemed a bit vague at first sight. To my point of view,  very generic statements such as «Add noise with size $\eta$» do not convey a clear point. I think the paper would benefit from focusing on the Gaussian noise injection, as it is the distribution that is used to demonstrate all the results.

Quality:
I had some trouble evaluating the soundness of the technical results because of the clarity issues mentioned above. Also, I am a bit uncomfortable with the way the authors demonstrate their statements. Some results (Lemma 1, Propositions 1 and 2, Corollary 1) are not provided with any formal proof. Some others (Theorems 2 and 3) are demonstrated only in the case of the linear regression. Even though the results presented in Appendix and existing works in the domain might be sufficient to provide simple proofs for these results, I feel like this should not be the reviewer’s (nor the reader’s for that matter) job to check for the soundness of the statements by combinations of different (rather technical) results . I think the authors should provide clear proofs for the result they state, even if those are very simple. As in the clarity section, I suggest providing clear statements with proofs at least for the regression case in the main paper and provide some leads for generalizing the results in the appendix.

On the experimental part, I felt that the point was rather not very convincing and was not sufficiently correlated with the results. This impression comes mostly from the fact that 1) the mean of the Lipschitz constants with and without noise injection do not seem to differ in a significant way and 2) low-dimensional intuitions are usually misleading when dealing with adversarial examples. This problem on the experimental part is not my main concern because this is a theory paper, but I think it would be better to either present more convincing results or simply no experiments at all.

Originality and Significance:
I am might not be fully aware of the literature but I think that studying adversarial training through the scope of algorithmic stability is a new idea. The proposed solution of injecting noise is not very surprising (looking at the literature on algorithmic stability and specifically on the link between differential privacy and algorithmic stability) but the authors have the merit of using this solution in a new problem. I think the overall idea has some merit and this paper could become an interesting first step toward a better analysis of adversarial training once the technical content is polished.

Some bibliographical notes: Here are some missing works that I think are worth mentioning.

On generalization of adversarial machine learning

[1] J. Khim, et al. ‘‘Adversarial risk bounds for binary classification via function transformation’’ 2018

[2] P. Awasthi, et al. “Adversarial Learning Guarantees for Linear Hypotheses and Neural Networks” 2020

[3] R. Pinot, et al. ‘‘On the robustness of randomized classifiers to adversarial examples’’ 2021

On using noise injection to stabilize the model against adversarial attacks

[4] B. Wang, et al. ‘‘ResNets Ensemble via the Feynman-Kac Formalism to Improve Natural and Robust Accuracies’’ 2019

[5] J. Cohen, et al. ‘‘Certified Adversarial Robustness via Randomized Smoothing’’ 2019

[6] N. Ford, et al. ‘‘Adversarial Examples Are a Natural Consequence of Test Error in Noise’’ 2019

[7] T. Weng, et al. ‘‘Evaluating the robustness of neural networks: an extreme value theory approach’’ 2018


**Time Spent Reviewing:**

Between 5 and 7 hours

---

> ### Author Response · Authors · 2021-08-06
> **Feedback to Reviewer 4**
>
> We appreciate your effort in reviewing our paper! We will try to improve the quality of this paper to make it more clear. Below are our comments towards your review:
>
> 1. Clarity:
>     1. Thanks for letting us know about your feeling about Lemma 1. It is the core result we used for the following theorems of algorithmic stability. We may wrap up the explanations and try to hide the technical details of this lemma from the main text.
>     2. We provided the results for other loss functions in Section D.3. We may consider writing a clear list to specify these.
>     3. For Algorithm 1, we may change it to Gaussian noise and add a remark mentioning that the noise can be more general.
> 2. Quality:
>     1. Theory: we provided rigorous proofs for all the lemmas and theorems in appendix Section D. Proposition 1 and 2 are some simple extensions from existing results.
>     2. Experiment: due to the space limit, we postponed most numerical experiments in the appendix. There are both simulations and real-data experiments in Section B and C.2 in the appendix. We will mention the extra experiments in the main text.
> 3. Missing works: thanks for providing a long list of missing references!

---

### Official Review · Reviewer_MT9T · 2021-07-18

**Rating:** 5
**Confidence:** 4

**Summary:**

The paper investigates the stability of adversarial training when using SGD/GD, motivated by the stability analysis of Bassily et al. (2020) and Hardt et al. (2016) in the standard learning model.
In order to improve stability, the authors suggest injecting noise into the weights and data
(this was already investigated, see for example - Adversarial weight perturbation helps robust
generalization, Neurips2020).
Moreover, theoretical analysis is also provided for Linear Regression, Logistic Regression and learning with hinge loss.


**Ethics Review Area:**

["I don’t know"]

**Limitations And Societal Impact:**

-

**Main Review:**

The main contributions are on the theoretical side.
My main concerns are about originality/novelty. The upper and lower bound on stability are adaptations of the claims of Bassily et al. (2020).
Moreover, methods of weight perturbation were already investigated empirically  (see for example - Adversarial weight perturbation helps robust generalization, Neurips2020), and some theoretical justifications were also provided in follow-up papers.

Moreover, there are glaring omissions in the literature review, especially on the theoretical side and regrading weights perturbation methods.

On the other hand, I think that the main research question is interesting and important.

**Time Spent Reviewing:**

3

---

> ### Author Response · Authors · 2021-08-06
> **Feedback to Reviewer 3**
>
> Thanks for reviewing our paper! We focused more on the algorithmic stability part for the literature review and thus did not include too much empirical literature in adversarial training or theoretical results in areas that are not closely related. Based on our knowledge, our aspect of algorithmic stability in noise-injected adversarial training is novel. For example, in [10], the theoretical justifications for random smoothing are on the existence of a robust model, but not how the noise injection works throughout the iterative training process or its algorithmic stability. We will also follow the suggestion of Reviewer 4 to add some more literature.
>
> [10] Cohen, Jeremy, Elan Rosenfeld, and Zico Kolter. "Certified adversarial robustness via randomized smoothing." International Conference on Machine Learning. PMLR, 2019.

---

### Official Review · Reviewer_ZiiQ · 2021-07-18

**Rating:** 7
**Confidence:** 3

**Summary:**

This paper provides the argument stability bound for adversarial training for linear models and 2-layer NNs with or without noise injection. The stability has a direct connection to the generalization bound. The contribution is mainly on the theoretical side. Motivated by the non-smoothness in AT which causes poor stability, the paper proposes to inject noise during training for both weight and data. Theory and simulation results justify noise injection on improving stability and shrinking generalization gap.

**Limitations And Societal Impact:**

The authors adequately addressed the limitations. As a theory paper, the potential negative societal impact of their work is little.

**Main Review:**

Pros:
- Provides the first lower bound and upper bound for uniform argument stability (UAS) for AT
- Considers numerical attack error in inner maximization which better models the real scenario
- Theoretically analyzes how noise injection improves the stability
- The paper is well written and easy to follow.

Cons:
- The theoretical results are limited to linear models (linear/logistic regression, smooth hinge loss) and two-layer NNs. Moreover, The UAS bounds without noise injection seem to be limited to convex and L-Lipschitz loss. Not sure whether it is a standard condition for UAS community.
- The connection of UAS bound to generalization error bound seems not that direct:
  - What are the proper conditions for Eqn. (3)? It seems that Eqn. (3) only relies on the argument stability but not any other properties of the model $f_\theta$ itself, which is a bit weird. Are there any assumptions here, e.g., $f_\theta$ is a linear model, $f_\theta$ is convex or $L$-Lipschitz?
  - Can Theorem 1 and Corollary 1 be extended to provide generalization bound for GD/SGD trained models? Do we need to further assume the tail distribution of $\sup \mathbb{E} ||\theta_1^{(T)} - \theta_2^{(T)}||$? Why do we have such generalization bound for noise injection (Proposition 2) regime but not for the normal regime?

Minor:
- What does $B$ stand for in Lemma 1?
- It would be better if the $x$-axis scales in Figure 1 could be the same to allow for direct comparison.
- I like the empirical evidence in C.2.1 that shows how noise injection shrinks the generalization gap in real-world data. Maybe move it to the main text?

I don't work in theory and didn't check the correctness of the proof. While it appears to me that UAS bounds provide a new perspective on analyzing the generalization and stability for AT, and the UAS bounds in this paper are technically nontrivial. I tend to support the acceptance but would adjust my evaluation based on theory reviewers.

**Time Spent Reviewing:**

3

---

> ### Author Response · Authors · 2021-08-06
> **Feedback to Reviewer 2**
>
> Thanks for reviewing our paper! Below are our comments toward the cons and minor issues in your review:
>
> 1. We acknowledge the big gap between the model studied in our paper and the DNN models used in real applications. The fact is that investigating the theoretical behavior of DNN training is a rather challenging task at the current stage. Most existing theoretical literature only consider one simple loss function (e.g., square loss [5,6]) under some simple data-generating models (e.g., linear model [5,6], mixed Gaussian model [7,8], or linear hyperplane in classification [9]) and can only deliver upper bound results.
> 2. Connection of UAS bound to the generalization error:
>     1. The conditions for eqn (3) are mentioned before this equation, i.e., line 121-122. This is a general result of connecting algorithmic stability to generalization, while the parameters in the UAS condition are determined by the data distribution, model, and loss functions.
>     2. Theorem 1 and Corollary 1 can provide the upper bound of the generalization error, but since these algorithmic stability bounds are already suboptimal, it only implies a suboptimal upper bound for the generalization error. There is no conclusion on the generalization error for the vanilla adversarial training. For example, [3] and [9] both apply adaptations to the training algorithm to get generalization results, and those are not the vanilla algorithm.
>
> In terms of the definition of B mentioned in the minor issues, it is the Lipschitz constant associated with the gradient of the loss, e.g. in the statement of Lemma 4 on P21 in the supplementary material. Thanks for the list of minor issues. We will update them in our revision.
>
> [1] Sinha, Aman, et al. "Certifying some distributional robustness with principled adversarial training." arXiv preprint arXiv:1710.10571 (2017).
>
> [2] Ba, Jimmy, et al. "Generalization of two-layer neural networks: An asymptotic viewpoint." International conference on learning representations. 2019.
>
> [3] Xing, Yue, Qifan Song, and Guang Cheng. "On the Generalization Properties of Adversarial Training." International Conference on Artificial Intelligence and Statistics. PMLR, 2021.
>
> [4] Rice, Leslie, Eric Wong, and Zico Kolter. "Overfitting in adversarially robust deep learning." International Conference on Machine Learning. PMLR, 2020.
>
> [5] Javanmard, Adel, Mahdi Soltanolkotabi, and Hamed Hassani. "Precise trade-offs in adversarial training for linear regression." Conference on Learning Theory. PMLR, 2020.
>
> [6] Xing, Yue, Ruizhi Zhang, and Guang Cheng. "Adversarially Robust Estimate and Risk Analysis in Linear Regression." International Conference on Artificial Intelligence and Statistics. PMLR, 2021.
>
> [7] Chen, Lin, et al. "More data can expand the generalization gap between adversarially robust and standard models." International Conference on Machine Learning. PMLR, 2020.
>
> [8] Dan, Chen, Yuting Wei, and Pradeep Ravikumar. "Sharp statistical guaratees for adversarially robust gaussian classification." International Conference on Machine Learning. PMLR, 2020.
>
> [9] Allen-Zhu, Zeyuan, and Yuanzhi Li. "Feature purification: How adversarial training performs robust deep learning." arXiv preprint arXiv:2005.10190 (2020).

---

> > ### Comment · Reviewer_ZiiQ · 2021-08-29
> > **Thanks for Response**
> >
> > I appreciate the detailed response and my major concerns are addressed.
> >
> > The paper provides new results on the algorithmic stability of adversarial training, which seems to be a novel and useful angle. Even though the DNN extension of current results is challenging, this stability viewpoint can inspire the community. The noise injection for improving stability has a clear motivation and theoretical backing.
> >
> > However, as pointed out by Reviewer 6caz, the presentation can be improved. Section 4.2 is too complex with various notations where the definitions are hard to find, and one has to refer to the appendices to understand the theorem statements. The authors may need to follow Reviewer 6caz's suggestions to improve clarity.
> >
> > Overall, I will keep my score unchanged given the contribution of this work even though there are clarity issues. I slightly increased my confidence from 2 to 3.

---

> > > ### Author Response · Authors · 2021-08-29
> > > **Thanks for the follow-up**
> > >
> > > We appreciate your effort in reviewing this paper and following up with our feedbacks! We will add the additional related references and improve the clarity of our paper based on the comments from Reviewer 6caz.

---

### Official Review · Reviewer_Sgsz · 2021-07-25

**Rating:** 7
**Confidence:** 4

**Summary:**

This work builds upon Bassily et al. and extends stability analysis to adversarial training algorithms. The authors use uniform algorithmic stability to derive lower and upper bounds on the robust accuracy.  Based on the theoretical analysis, they argued that non-differentiability (or gradient-masking) reduces the stability of adversarial training, which leads to the significant gap between training and testing robust accuracy. To reduce the generalization gap, the authors suggest using noise injection into the parameters and the input of the models.

**Limitations And Societal Impact:**

### Limitations:
- Limitations of the work are discussed.

### Societal impact:
- The authors did not discuss societal impact.

**Main Review:**

### Originality:
- The work extends in a novel way uniform stability analysis to adversarial training algorithms.
- The authors provide bounds in case of an imperfect attack, which is the case in all applications of adversarial training.

### Quality:
- The paper gives new insights into adversarial training and in a way explains catastrophic overfitting with weak adversaries.
- The claims seem to be theoretically sound and correct.

### Clarity:
- The paper is well motivated and superbly written.

### Significance:
- The developed insights may lead to the development of new robust training algorithms.

### Questions to the authors:
1) The bounds for SGD does not depend on the batch size, which seems to be counterintuitive. Can the authors clarify that?
2) Is the input noise injection equivalent to random start for PGD adversarial training? If it is not, what are the differences?
3) What do the authors mean by vanishing initialization in Figure 1 as it is not clear from the context?
4) In appendix A remark 4, the authors state "adversarial training prefers a smaller number of steps to reduce the
corresponding UAS upper bound". It seems to be correct for training with weak adversaries but incorrect for training with strong adversaries, since robust training with a strong adversary requires many more iterations. Can the authors clarify if I am wrong?
5) In appendix C2.1, noise injection and data augmentation reduce adversarial test accuracy. Can the authors comment on this result?

**Time Spent Reviewing:**

6

---

> ### Author Response · Authors · 2021-08-06
> **Feedback to Reviewer 1**
>
> We appreciate your efforts in reviewing this paper! Below are some feedback on your questions:
>
> 1. We follow the definition in Bassily et al. (2020) to take the batch size as 1. The SGD result in Theorem 3 (the convergence of adversarial training) will not change as the batch size. The proof of Theorem 3 is a common approach for the convergence of SGD, e.g., Theorem 2 of [1]. When taking expectations, SGD and GD become the same.
> 2. The purpose of input noise is different from that of the random start for PGD attack. The former is to have a better chance to find a good attack, which is expected to converge to the fixed “optimal” attack. In contrast, the input noise injection aims to skip the non-differentiable points of the loss function and is never fixed.
> 3. We follow [2] and [3] for the definition of vanishing initialization: asymptotically, if the data dimension, the number of hidden nodes, and the number of samples increase to infinite, then the initialization will converge to zero. We will add more explanations in the revision.
> 4. We claim that a smaller number of optimization steps leads to better stability, while for strong adversaries, a large number of steps is necessary for convergence. This is not a contradiction but a trade-off between the convergence of adversarial training and algorithmic stability. As studied by [4], adversarial training is likely to overfit, thus how to balance the optimization convergence (of strong attack) and the algorithmic stability is an important yet not solved problem.
> 5. The aim of noise injection is not to improve the final testing performance but to reduce the generalization gap (without sacrificing the testing performance). From this aspect, the idea implied from our theorems in simple models matches Table 2 in Appendix C2.1. In Table 2, the difference between the adv test acc before/after injecting noise for AT is not significant, and there are many potential issues affecting this adv test acc.
>
> [1] Sinha, Aman, et al. "Certifying some distributional robustness with principled adversarial training." arXiv preprint arXiv:1710.10571 (2017).
>
> [2] Ba, Jimmy, et al. "Generalization of two-layer neural networks: An asymptotic viewpoint." International conference on learning representations. 2019.
>
> [3] Xing, Yue, Qifan Song, and Guang Cheng. "On the Generalization Properties of Adversarial Training." International Conference on Artificial Intelligence and Statistics. PMLR, 2021.
>
> [4] Rice, L. et. al., Overfitting in adversarially robust deep learning. arXiv preprint arXiv:2002.11569 (2020)

---

### Decision · Program_Chairs · 2021-09-27

**Decision:**

Accept (Poster)

**Comment:**

The large generalization gap of adversarial training is a central problem of adversarial robustness. This paper makes the first attempt to study this problem from an algorithmic stability point of view. The authors establish upper and lower bounds for robust accuracy of adversarial training and investigate the causes of large generalization gap. In addition, the paper proposes a noise-injection method to improve the algorithmic stability and threfore reduces the generalization gap of adversarial training. Overall, this is a nice contribution to NeurIPS. It provides insightful understanding to a key problem of adversarial training. On the other side, the paper mainly contains theoretial analysis. More experimental study would benefit more readers. I recommend accept.